# EFFICIENT AND SCALABLE MARL FROM IMAGES BY TRUST-REGION AUTOENCODERS

## ABSTRACT

Vision-based multi-agent reinforcement learning (MARL) suffers from poor sample efficiency, limiting its practicality in real-world systems. Representation learning with auxiliary tasks can enhance efficiency; however, existing methods, including contrastive learning, often require the careful design of a similarity function and increase architectural complexity. In contrast, reconstruction-based methods that utilize autoencoders are simple and effective for representation learning, yet remain underexplored in MARL. We revisit this direction and identify unstable representation updates as a key challenge that limits its sample efficiency and stability in MARL. To address this challenge, we propose the Multi-agent Trust Region Variational Autoencoder (MA-TRVAE), which stabilizes latent representations by constraining updates within a trust region. Combined with a state-of-the-art MARL algorithm, MA-TRVAE improves sample efficiency, stability, and scalability in vision-based multi-agent control tasks. Experiments demonstrate that this simple approach not only outperforms prior vision-based MARL methods but also MARL algorithms trained with proprioceptive state. Furthermore, our method can scale up to more agents with only slight performance degradation, while being more computationally efficient than the underlying MARL algorithm.

## 1 INTRODUCTION

Recent advances in multi-agent reinforcement learning (MARL) have demonstrated its potential in sequential decision-making domains such as autonomous driving (Dinneweth et al., 2022), robotics (Orr & Dutta, 2023), and large-scale control (Ma et al., 2024), where multiple agents must learn to coordinate from partial and local observations. Nevertheless, MARL methods remain notoriously sample-inefficient, often requiring millions of interactions with the environment before converging to effective policies. This issue becomes even more pronounced in vision-based settings, making learning slower, less scalable, and far less practical for real-world systems, where cameras are a convenient and inexpensive way to perceive the environment.

The sample efficiency of reinforcement learning with visual input has been extensively studied for single-agent systems. The most popular strategy is to learn a good lower-dimensional representation of the high-dimensional input. This approach is based on the hypothesis: *learning a policy with a semantically meaningful low-dimensional representation of the visual input is significantly more sample efficient.* Different ways of obtaining such representations have been explored, including but not limited to (i) reconstruction-based representation learning (Yarats et al., 2021), (ii) contrastive learning

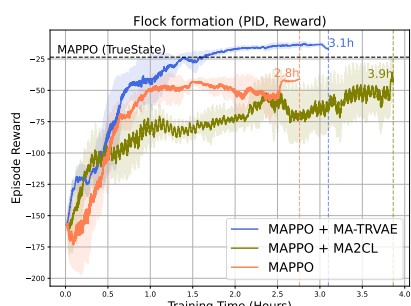

Figure 1: Episode reward vs. training time. *We evaluate MA-TRVAE +MAPPO on a multi-agent quadcopter task, where four drones need to form a flock with ego-centric views. Masked attentive contrastive learning (MA2CL) is a state-of-the-art representation learning method on this benchmark, yet it shows an unstable performance. MA-TRVAE is more computationally efficient than MAPPO and 20% faster than MA2CL, with the final episodic reward even higher than MAPPO trained with proprioceptive state.*

(Laskin et al., 2020; Stooke et al., 2021), and (iii) world models (Ha & Schmidhuber, 2018b; Hafner et al., 2019a; 2021; 2025).

In the MARL literature, Song et al. (2023) utilize contrastive learning and an attention module (Vaswani et al., 2017) to learn a representation that can capture temporal and agent-level information, while Feng et al. (2025) introduce a transition-informed framework to learn an attention-based world model that captures the dynamics of the multi-agent systems. While those previous works, which we discuss in more detail in appendix A, improve the sample efficiency of vision-based MARL by using a complex contrastive learning algorithm and attentive modules that scale quadratically with respect to input length, we tackle this challenge from a practical perspective: *how to design a method that is both simple and computationally efficient?* This leads us to reconstruction-based representation learning, which is simple to implement and computationally efficient as it uses convolutional neural networks (CNNs) (LeCun et al., 1989) that scale linearly with the image size. Moreover, it is a general method for representation learning from images. Unlike contrastive learning, it does not require designing positive and negative samples, nor a similarity function. Additionally, it does not rely on learning the underlying dynamic model of the system, which can be challenging for multi-agent systems that are only partially observable. Surprisingly, this approach is underexplored in MARL, to the best of our knowledge.

Following previous work in the single-agent setting (Yarats et al., 2021), we start by integrating *variational autoencoders* (VAEs) with *multi-agent proximal policy optimization* (MAPPO) (Yu et al., 2022) and confirm that looser constraints in the latent space of the VAE improve the stability and task performance for MARL, as it is observed in the single-agent case. However, a small constraint in the latent space can destabilize the representation update. That is, the representation of the same observation can change dramatically from one encoder update step to another, and the policy may perceive the same observation differently, thus harming sample efficiency. This issue is amplified in MARL, where observations from different agents are diverse.

On the other hand, trust region methods are widely applied in reinforcement learning since they improve the stability of the policy update (Schulman et al., 2015; 2017b). Instead of applying them to policy learning in MARL, as in prior work (Yu et al., 2022), we adapt them to representation learning to address the instability of representations, which arises from the lack of constraints in the latent space. To that end, we propose Multi-agent Trust Region VAE (MA-TRVAE), which constrains updates to keep new representations close to previous ones. Combined with MAPPO, we show that MA-TRVAE improves sample efficiency and stability across multiple vision-based control tasks, while being computationally more efficient than MAPPO and 20% faster than a state-of-the-art method. Furthermore, it even outperforms MAPPO with proprioceptive state for the final return as shown in figure 1. We then scale our methods to up to 7 agents and observe only a modest decline in performance, while state-of-the-art baseline methods degrade significantly.

In summary, our main contributions are (i) we study reconstruction-based representation learning in MARL and propose MA-TRVAE, a novel framework that adapts trust region methods to stabilize representation learning in multi-agent settings, (ii) we demonstrate that MA-TRVAE significantly improves sample efficiency and stability in vision-based MARL tasks, outperforming strong baselines including MAPPO with proprioceptive states for the final return, while being more computational efficient than MAPPO with visual input, and (iii) we show that MA-TRVAE scales effectively to larger numbers of agents, maintaining strong performance while competing methods degrade more rapidly. [1]

## 2 PROBLEM SETTING AND BACKGROUND

This section defines the problem setting and provides the required background on MAPPO, which we use as our baseline algorithm, and representation learning.

### 2.1 PROBLEM SETTING

We consider a cooperative MARL setting with partial observability for each agent, which we model as a decentralized-partially observable Markov decision process (Dec-POMDP) (Oliehoek & Am-

---

[1]See https://sites.google.com/view/stablerepresentation for videos of the experiments and code.

ato, 2016). A Dec-POMDP is defined as $\langle \mathcal{N}, \mathcal{O}, \mathcal{A}, R, P, \gamma \rangle$, where $\mathcal{N} = \{1, ..., N\}$ is the finite set of $N$ agents, $\mathcal{O} = \prod_{i=1}^{N} \mathcal{O}^i$ is the joint observation space, which is the Cartesian product of the local observation spaces $\mathcal{O}^i \subseteq \mathbb{R}^{H \times W \times C}$ (since we consider vision-based observation, where H, W, and C are height, width, and number of channels of the image), $\mathcal{A} = \prod_{i=1}^{N} \mathcal{A}^i$ is the joint action space, composed of the local action spaces $\mathcal{A}^i \subseteq \mathbb{R}^b$, $R : \mathcal{O} \times \mathcal{A} \to \mathbb{R}$ is the joint reward function, $P : \mathcal{O} \times \mathcal{A} \times \mathcal{O} \to [0, 1]$ is the state transition probability function, and $\gamma \in [0, 1)$ is the discount factor. At each time step $t \in \mathbb{N}$, each agent receives a local observation $o_t^i \in \mathcal{O}^i$ and takes an action $a_t^i$ according to its policy $\pi^i : \mathcal{O}^i \to \mathcal{A}^i$. The next set of observations $o_{t+1}$ is updated based on the transition probability function $P$, and the entire team receives a joint reward $R(o_t, a_t)$. The goal is to maximize the expected cumulative joint reward over a finite or infinite number of steps,

$$\max_{\pi} \mathbb{E}_{\boldsymbol{\pi}} \left[ \sum_{t=0}^{T} \gamma^t R(o_t, a_t) \right]. \tag{1}$$

## 2.2 MAPPO

In MARL, a widely adopted framework to stabilize learning, which is also used by MAPPO, is centralized training with decentralized execution (CTDE), where agents have access to the global state and other agents' actions during training and use only local observations during execution. In such methods (Lowe et al., 2017; Rashid et al., 2018; Yu et al., 2022; Kuba et al., 2021), an encoder from the decentralized part of the algorithm, such as the actor in policy gradient methods, produces the observation representations, $z_t^i$. Let $g_\phi$ denote the encoder parameterized by $\phi$. Then, the observation representation can be expressed as $z_t^i = g_\phi(o_t^i)$. This latent variable is fed into either the policy network or the value network, which allows us to calculate the losses and backpropagate the gradients to optimize the networks.

In this paper, we use multi-agent proximal policy optimization (MAPPO) as the base MARL algorithm. MAPPO (Yu et al., 2022) is an extension of PPO to MARL. The representation encoder processes the observation of the agent, and the policy network generates an action based on the representation it produces. MAPPO updates the parameters using the aggregated trajectories of all agents collected from the current policy. At iteration $k + 1$, similar to equation 7, the policy parameters $\theta_{k+1}$ are optimized by maximizing the clipped objective,

$$J_\pi(\theta) = \sum_{i=1}^{N} \mathbb{E}_{o \sim \mathcal{D}, a \sim \pi_{\theta_k}} \left[ \min \left( r_k(\theta) A_{\pi_{\theta_k}}(o, a), \text{clip}(r_k(\theta), 1 \pm \tau) A_{\pi_{\theta_k}}(o, a) \right) \right], \tag{2}$$

where $r_k(\theta) = \frac{\pi_\theta(a^i|o^i)}{\pi_{\theta_k}(a^i|o^i)}$, $\theta_k$ is the policy parameter at iteration $k$, and $N$ denotes the number of agents. During training, MAPPO employs a centralized critic network, which uses the joint observation to estimate a value function, $V_w$. This centralized value function provides a more stable and informative advantage estimate for updating each agent's policy. The critic parameters are trained to minimize the temporal-difference error using aggregated trajectories from all the agents. The critic is updated separately using the objective

$$J_V(w) = -\sum_{i=1}^{N} \mathbb{E}_{o \sim \mathcal{D}} \left[ \left( V_w(o_t) - \hat{R}_t \right)^2 \right], \tag{3}$$

where $\hat{R}_t$ is the discounted reward.

## 2.3 Reconstruction-based representation learning and Autoencoders

Reconstruction is a self-supervised learning method to learn low-dimensional representations from images. It utilizes an autoencoder, typically consisting of a convolutional encoder $g_\phi$ and a deconvolutional decoder $f_\psi$, to first map an image $o$ to a low-dimensional latent vector $z$, then reconstruct the image $o$ from the latent vector $z$. Early work uses a deterministic autoencoder (Hinton & Salakhutdinov, 2006; Vincent et al., 2008), where the latent vector $z$ is deterministic. Variational autoencoders (VAE) are introduced by (Kingma & Welling, 2013) and greatly improve the representation capacity. In VAEs, the latent vector $z$ is a random variable and we assume a posterior

distribution $q_\phi(\boldsymbol{z} \mid \boldsymbol{o})$ over $\boldsymbol{z}$ given the observation $\boldsymbol{o}$ as well as a prior distribution $p(\boldsymbol{z})$ over $\boldsymbol{z}$, which normally is a standard multivariate Gaussian $\mathcal{N}(0, I)$. We can further encourage the disentanglement of the latent representation by setting a large constraint in the latent space using a $\beta$-VAE (Higgins et al., 2017), whose objective function is

$$\mathcal{L}_{\beta\text{-VAE}}(\psi, \phi) = \mathbb{E}_{\boldsymbol{o} \sim \mathcal{D}} \Big[ \mathbb{E}_{\boldsymbol{z} \sim q_\phi(\boldsymbol{z}|\boldsymbol{o})} \big[ \log p_\psi(\boldsymbol{o} \mid \boldsymbol{z}) \big] \; - \; \beta \, D_{\mathrm{KL}} \big( q_\phi(\boldsymbol{z} \mid \boldsymbol{o} \, \| \, p(\boldsymbol{z})) \big) \Big], \qquad (4)$$

where $\beta \in \mathbb{R}$ is a coefficient that controls the regularization on the latent space, i.e., the *Kullback-Leibler (KL)* divergence between the posterior and the prior. A larger $\beta$ enforces more constraints and tends to disentangle the latent vector more. However, Yarats et al. (2021) show that a smaller $\beta$ in equation 4 can improve the downstream single-agent RL performance when jointly training the autoencoder with an off-policy RL algorithm.

## 3 RECONSTRUCTION-BASED REPRESENTATION LEARNING FOR MARL

Following the work by (Yarats et al., 2021), we integrate a $\beta$-VAE with MAPPO and jointly optimize the autoencoder and MARL networks, including local actors and a shared critic. We specifically choose MAPPO over other off-policy MARL algorithms, such as MADDPG (Lowe et al., 2017), primarily for practical reasons. Although off-policy algorithms tend to be more sample-efficient by reusing samples from the replay buffer, their training cannot be as easily parallelized as on-policy algorithms like MAPPO, which enables the efficient use of computing resources and, in turn, results in better computational efficiency. We describe the detailed method in section 3.1 and show its surprising effect on improving sample efficiency for vision-based multi-agent control tasks by simply reducing the regularization on the latent space of the $\beta$-VAE. Although this simple application of $\beta$-VAE already improves the sample efficiency of MAPPO by a large margin, we show in section 3.2 that representations learned from such a method are not stable across steps. This motivates the development of MA-TRVAE in section 4.

### 3.1 SURPRISING EFFECT OF $\beta$-VAE ON MARL

We evaluate the performance of MAPPO with $\beta$-VAE on two multi-agent quadcopter control tasks, with PID action setting as detailed in section 5. In these tasks, four agents must cooperate to achieve different goals using visual input.

We first pretrain a shared convolutional encoder $g_\phi$ and a deconvolutional decoder $f_\psi$ using the objective $\mathcal{L}_{\beta-\text{VAE}}$ given in equation 4 on trajectories collected under a random policy. Subsequently, the actor and centralized critic are trained for $T$ steps with latent states $\boldsymbol{z}_t^i$ and $[\boldsymbol{z}_t^1, ..., \boldsymbol{z}_t^N]$ respectively, where $\boldsymbol{z}_t^i \sim g_\phi(\boldsymbol{o}_t^i)$, while the encoder $g_\phi$ remains fixed. The resulting policy is then deployed to all agents and used to gather new trajectories. Using the trajectories collected from all agents with the current policy, the shared encoder $g_\phi$ is updated using gradients from the actor loss as given in equation 2, critic loss as given in equation 3, and the $\beta$-VAE loss as given in equation 4, unlike Yarats et al. (2021), who exclude the actor's gradient when updating the encoder. We update the shared encoder after every policy update together with the decoder, since it is empirically shown by Yarats et al. (2021) that there is a positive correlation between the encoder updating frequency and the performance of the RL agent. This process of joint training of $\beta$-VAE and actor–critic network is repeated until convergence. Also, following the observation by Yarats et al. (2021) that a small value of $\beta$ improves the stability and task performance, we conduct the experiment for varying $\beta$ values and confirm that a less constrained latent space also enhances performance in MARL, as shown in figure 2.

Interestingly, we see a large gap between different values of $\beta$, which is not observed in single-agent systems (Yarats et al., 2021). Specifically, MAPPO that employs a $\beta$-VAE with $\beta$ values of 1 and 0.1 performs substantially worse than MAPPO without representation learning, whereas MAPPO with smaller values of $\beta$ yields a surprisingly large performance gain. This suggests that strong regularization on the latent space actually hurts the representation learning in multi-agent systems. We hypothesize that this effect arises from the heterogeneity of observations across agents. Unlike single-agent systems, where an agent receives a fixed view and global observation of the system, each agent in multi-agent systems perceives the environment partially from its own ego-centric view. The autoencoder must compress this diverse high-dimensional distribution of observations into a

low-dimensional latent space. When the latent space is over-regularized, it may lack the expressive capacity required to capture such complexity in the observation.

Another trend we observe in figure 2 concerns training stability. Specifically, the variance of episodic rewards decreases as the value of $\beta$ becomes smaller. This aligns with findings from single-agent systems (Yarats et al., 2021), which reported that the stochastic nature of a $\beta$-VAE damages the performance of the RL agent. Indeed, a stochastic representation introduces more variance for the underlying RL policy. However, when the latent space is more stochastic, namely, when $\beta = \{1, 0.1\}$, the variance in MARL agents' performance is even smaller than with an extremely small $\beta$ value, e.g., $10^{-7}$. This seemingly contradictory observation motivates us to further investigate instability in MARL performance. In the next section, we show that the stochasticity in the representation is not the sole factor causing instability; weak constraints on the latent space also contribute to this issue.

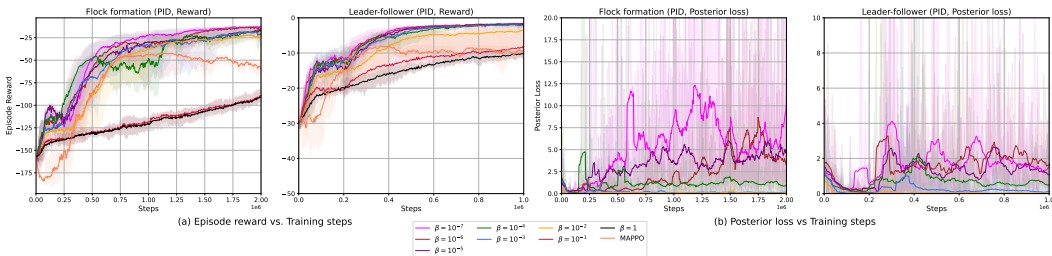

Figure 2: Episode reward and posterior loss vs. training steps plot for flock and leader-follower tasks, with PID action setting, using different $\beta$ values. *We see that, as $\beta$ decreases, the performance enhances. However, it leads to inconsistent representations, which can be observed from the increased posterior loss of vanilla VAE.*

## 3.2 INSTABILITY OF REPRESENTATIONS

We investigate another source of instability by understanding how the representation changes over steps. Recall that the encoder and actor-critic are updated alternately for each policy update step. This means that the representation used for policy learning changes between consecutive steps. If the latent space shifts drastically after an encoder update, the representation for the same observation can vary significantly. This undermines the sample efficiency of the underlying MARL agents, since the policy may perceive the same observation inconsistently and must remap them to the same optimal action. To study this effect, we track how the posterior of the latent vector $z$ for a fixed observation changes across steps, and hypothesize that weaker constraints on the latent space should lead to a large posterior deviation between two successive steps.

In figure 2, we plot the KL divergence between the posteriors of two successive steps for the same observation throughout training. When the value of $\beta$ is large, the divergence is close to zero. This is expected, since the large value of $\beta$ pushes the posterior towards $\mathcal{N}(0, I)$ strongly. By contrast, with weaker constraints (smaller $\beta$), the divergence grows faster, suggesting the representation for the same observation changes more across updates. Intuitively, this means that the representation of the same observation, instead of being concentrated in a close latent region, is more likely to scatter around the latent space. This investigation explains why the MARL agents' performance is more stable when using a large value of $\beta$, as stronger constraints on the latent space mitigate the representation drift.

Building on this discovery, a natural question arises: *can we further improve the stability of representations when using a small value of $\beta$?* Achieving this should further enhance the sample efficiency of the MARL agents. This motivates the use of trust region optimization. We discuss in the next section how to adapt this technique to stabilize representation learning with a small value of $\beta$, while preserving the computational efficiency of the $\beta$-VAE.

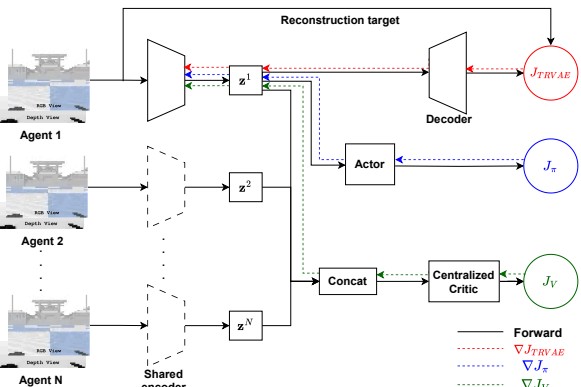
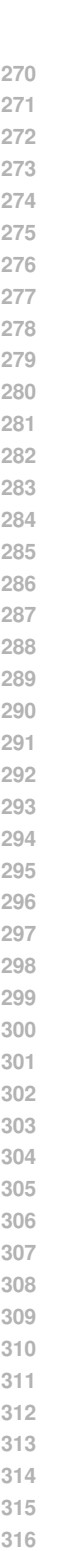

Figure 3: Schematic diagram of the proposed MA-TRVAE architecture. *The dotted encoder boxes represent deep copies of a single shared encoder across agents. The parameter update paths are illustrative—in implementation, the encoder is shared and updated with the gradients from the reconstruction $J_{TRVAE}$, value fuction $J_V$, and policy $J_\pi$ objectives as given in equation 6, equation 3, and equation 2 respectively.*

## 4 TRUST-REGION AUTOENCODER FOR MARL

To enhance stability of representation in MARL, this section introduces the trust-region variational autoencoder for MARL, which we refer to as MA-TRVAE.

Trust region methods are widely used to regularize the policy update in RL. The same idea can be directly applied to representation learning in MARL, where we update the posterior of the latent after each policy update. Formally, the application of a trust region method can be defined as a constrained optimization problem given the objective function in equation 4,

$$
\begin{aligned}
\max_{\psi,\phi} \quad & \mathcal{L}_{\beta-\text{VAE}} \\
\text{s.t.} \quad & \mathbb{E}_t\big[D_{KL}\big(q_\phi(\mathbf{z}_t|\mathbf{o}_t) \,\|\, q_{\phi_{old}}(\mathbf{z}_t|\mathbf{o}_t)\big)\big] \leq \delta,
\end{aligned}
\tag{5}
$$

with constraint $\delta \in \mathbb{R}$, and $q_{\phi_{\text{old}}}$ the posterior inferred by the encoder from the previous step. Although both the objective function and the constraint are based on the expected value, which can be approximated by a Monte Carlo estimator using observations collected by agents, solving this constrained optimization problem requires optimization techniques beyond standard backpropagation. For example, TRPO (Schulman et al., 2015) relies on a conjugate gradient method, a second-order optimization approach. This additional complexity makes the implementation more challenging and the method itself more computationally demanding, both of which can hinder the application of MA-TRVAE. To avoid this, we instead enforce the trust region constraint by introducing a penalty term into the objective function of the $\beta$-VAE, yielding an unconstrained optimization problem with a surrogate

$$
\begin{aligned}
\mathcal{L}_{\text{TRVAE}} = \mathbb{E}_{\mathbf{o}_t \sim \mathcal{D}} \big[ & \mathbb{E}_{\mathbf{z}_t \sim q_\phi(\mathbf{z}_t|\mathbf{o}_t)} \left[\log p_\psi(\mathbf{o}_t \mid \mathbf{z}_t)\right] \\
& - \beta_1 D_{\text{KL}}(q_\phi(\mathbf{z}_t \mid \mathbf{o}_t) \,\|\, p(\mathbf{z}_t)) \\
& - \beta_2 D_{\text{KL}}(q_\phi(\mathbf{z}_t|\mathbf{o}_t) \,\|\, q_{\phi_{\text{old}}}(\mathbf{z}_t|\mathbf{o}_t)) \big],
\end{aligned}
\tag{6}
$$

where $\beta_2 \in \mathbb{R}$ is a coefficient controlling the strength of the trust region penalty and $\beta_1$ corresponds to $\beta$ in the original $\beta$-VAE. With this objective function, MA-TRVAE constrains the updates of the autoencoder within a trust region, while remaining computationally efficient and easy to implement with backpropagation. We summarize MA-TRVAE in figure 3.

## 5 EXPERIMENTS

In this section, we describe the environment and experiments used to evaluate our proposed method. As we focus on vision-based observations, MARL environments that support such observations are needed, which are currently rare. Therefore, we consider the multi-agent quadcopter control (MAQC) (Panerati et al., 2021) environment for evaluating MA-TRVAE. Within this setting, we aim to verify three main hypotheses regarding MA-TRVAE: (i) Does the penalty term in equation 6 mitigate representation drift and thereby improve sample efficiency? (ii) How well does the method scale with the number of agents? (iii) Is our method more computationally efficient compared to prior approaches? Additionally, we recognize the generality of the trust-region VAE introduced in equation 6 and conduct experiments to demonstrate its impact in single-agent systems. We refer interested readers to appendix G for results and discussion.

### 5.1 EXPERIMENT SETUP AND BASELINES

**Multi-agent quadcopter control.** Each agent in the MAQC environment receives an RGBD video frame $\in \mathbb{R}^{64 \times 48 \times 4}$ as an observation. The observations are captured from a camera mounted on the drone toward the positive direction of the local $x$-axis. The MAQC environment also supports state-based observations, providing an observation vector that includes drones' positions, quaternions, linear velocities, angular velocities, and motor speeds. The environment provides three action settings: PID, DYN, and RPM, arranged in order of increasing difficulty. The RPM action setting enables agents to directly command motor speeds. In contrast, in the DYN action setting, agents produce torque values that determine motor speeds. Finally, the PID mode enables agents to output control inputs to a PID controller, which then calculates the appropriate motor speeds. In this paper, we evaluate our work in two cooperative multi-agent tasks, `flock` and `leader-follower`, in PID, DYN, and RPM action settings with 4 agents, except for the scalability experiment. We run all `flock` experiments for 2 million training steps and run all `leader-follower` experiments for 1 million training steps. More details about the tasks are provided in appendix C.

**Baselines.** We compare MA-TRVAE (with $\beta_1 = 10^{-7}$ and $\beta_2 = 10^{-6}$) to different versions of MAPPO (Yu et al., 2022): (i) the standard MAPPO algorithm, to demonstrate the performance gains through the VAE with trust-region, (ii) MAPPO with a vanilla VAE ($\beta = 10^{-7}$) (Kingma & Welling, 2022; Higgins et al., 2017), to investigate the impact of the trust-region itself, (iii) MAPPO with deterministic AE, which improves sample efficiency in single-agent systems (Yarats et al., 2021), (iv) MAPPO with state observations, which we call MAPPO(TrueState), and (v) MAPPO with masked attentive contrastive learning (MA2CL) (Song et al., 2023), which has reported state-of-the-art performance in the MAQC environment. For each algorithm and task, we independently run five experiments with random seeds to obtain the mean and standard deviation of various evaluation metrics, such as episode rewards and posterior loss. The hyperparameters are kept consistent with those used in the original papers for a fair comparison, and remaining the same across all experiments. More details about the hyperparameters can be found in appendix E.

### 5.2 RESULTS AND DISCUSSION

Having introduced the environment and baselines, we now verify the three hypotheses.

**Stability of representation.** We conduct an empirical study to show the effectiveness of the trust-region constraint in stabilizing the representation learning. The empirical study is conducted in both `flock` and `leader-follower` tasks, with PID action setting, to ensure consistency with the empirical study presented in section 3. As shown in figure 4, when a trust-region constraint is introduced to the loss function, the consistency in representations across the training steps is improved.

**Sample efficiency and stability.** To verify that the improved stability in representations translates into higher sample efficiency and stability in MARL, we evaluate the mean and standard deviations of the episode reward. From figure 5 and table 1, we first of all observe that MA-TRVAE outperforms all baselines in terms of mean episode reward across all tasks and action settings, even MAPPO with true states. Moreover, the variance is the lowest in five out of six cases. These results show that MA-TRVAE can reliably learn high-performing policies, making a case for the increased stability through the trust-region approach. Moreover, in terms of sample efficiency, we observe that only MAPPO

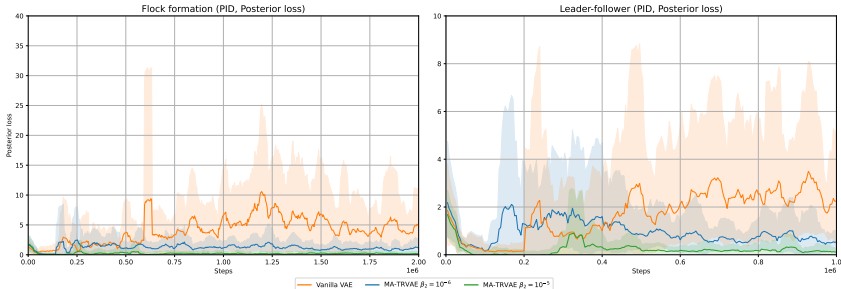

Figure 4: Posterior loss vs. training steps plot for flock and leader-follower tasks, with PID action setting, comparing vanilla VAE and MA-TRVAE. *We see that, by introducing a trust-region constraint, representations are becoming stable, which can be observed from the decreasing posterior loss throughout training.*

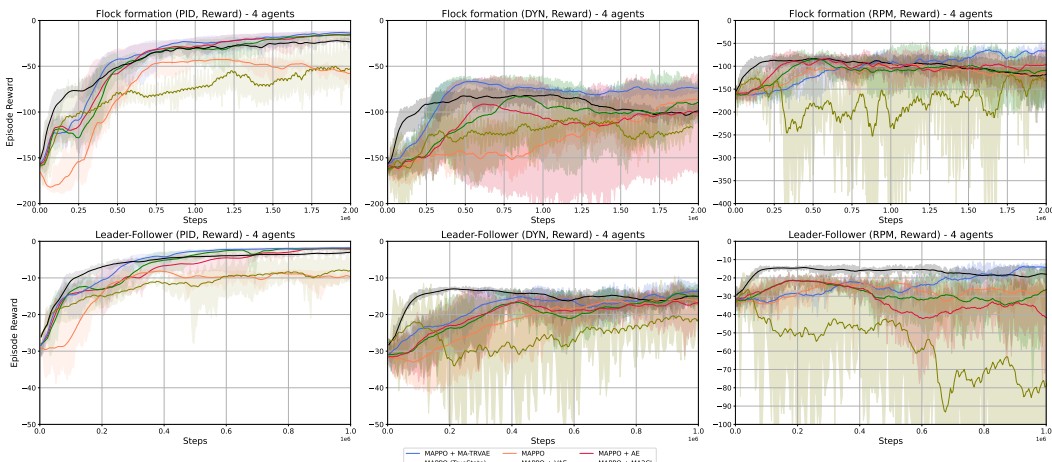

Figure 5: Episode reward vs. training steps for flock and leader-follower tasks, with PID, DYN, and RPM action settings. *We can observe that MA-TRVAE is performing better compared to other baselines in terms of better sample efficiency, stability, and low variance. It should be noted that from left to right, the difficulty of the task increases, which makes other algorithms unstable.*

with true states shows a faster performance increase. This is natural, as in that case, no learning of a lower-dimensional representation is required. Nevertheless, the results show a significant gain in sample efficiency through MA-TRVAE compared to all baselines that learn from visual input. Thus, through these experiments, we can verify the first hypothesis. To further evaluate the stability of MA-TRVAE, we provide additional experiments with noisy observations in appendix F.3.

**Scalability.** Next, we evaluate MA-TRVAE with 5, 6, and 7 agents to investigate its scalability. As shown in figure 6, MA-TRVAE maintains strong performance while remaining sample-efficient and stable compared to all baselines. Notably, MA2CL suffers a substantial performance drop when scaling from 4 to 5 agents (see the first column in figure 5 and 6), whereas other reconstruction-based methods show only minor changes. As the number of agents increases further, MA-TRVAE exhibits only a slight decline in performance, while the other reconstruction-based baselines degrade significantly, showing the effectiveness of our trust region regularization. These results (more can be found in appendix F.2) verify the second hypothesis, demonstrating the potential of MA-TRVAE for large-scale multi-agent control tasks with vision-based observations.

**Computational efficiency.** Lastly, we conduct experiments to understand how fast MA-TRVAE trains compared to the baselines. These experiments are performed in both `flock` and `leader-follower` tasks, with PID action setting on the same hardware configuration (see details in appendix E). We plot the episodic return against the running time for each method in figure 1 (complete results can be found in appendix F.1). As MA-TRVAE employs a simple reconstruc-

Table 1: Mean and standard deviation of the final episode reward. *MA-TRVAE outperforms the baselines in all tasks and settings.*

| | MAPPO+TRVAE | MAPPO+AE | MAPPO+VAE | MAPPO+MA2CL | MAPPO | MAPPO (TrueState) |
|---|---|---|---|---|---|---|
| **Flock-PID** | **-13.73 ± 0.96** | -16.02 ± 2.47 | -18.45 ± 3.09 | -52.71 ± 10.00 | -55.25 ± 2.80 | -22.90 ± **0.96** |
| **Flock-DYN** | **-74.69 ± 3.10** | -101.39 ± 25.01 | -93.24 ± 13.27 | -121.24 ± 4.82 | -89.10 ± 7.55 | -101.34 ± 3.30 |
| **Flock-RPM** | **-69.70 ± 4.03** | -100.45 ± 12.34 | -111.07 ± 24.49 | -135.82 ± 20.11 | -117.96 ± 10.68 | -118.62 ± 23.27 |
| **Leader-PID** | **-1.81 ± 0.09** | -2.09 ± 0.20 | -2.18 ± 0.29 | -8.24 ± 1.79 | -9.62 ± 0.82 | -3.24 ± 0.17 |
| **Leader-DYN** | **-14.09 ± 1.49** | -16.85 ± 2.73 | -14.69 ± 1.16 | -20.91 ± 2.93 | -16.01 ± **0.45** | -15.42 ± 1.10 |
| **Leader-RPM** | **-14.10 ± 0.57** | -36.63 ± 7.24 | -30.04 ± 8.98 | -79.59 ± 34.91 | -27.51 ± 2.52 | -18.63 ± 2.28 |

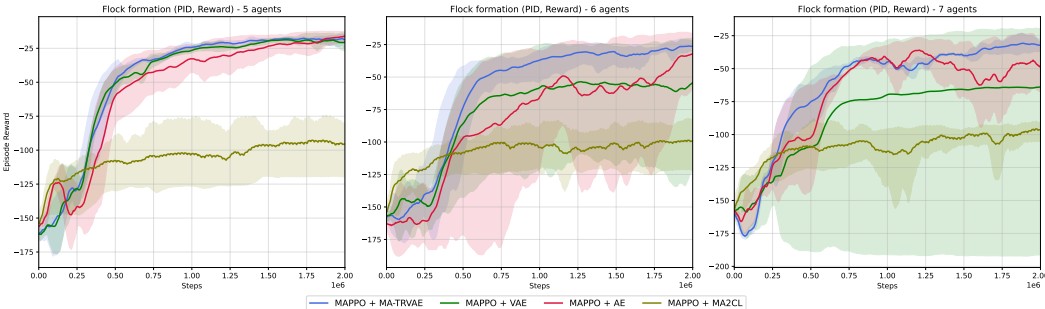

Figure 6: Episode reward vs. training steps for flock formation task, with PID action setting, for increasing number of agents. *We can observe that MA-TRVAE scales well with the number of agents, with a stable learning curve, outperforming the baselines*

tion method, its training time increses by only 10% over MAPPO thanks to the light computational cost of CNNs. MA2CL utilizes an attention module that requires more training time, to be precise, around 20% more training time than MA-TRVAE, while achieving lower performance and being less stable. This result also verifies our third hypothesis.

# 6 CONCLUSION AND LIMITATIONS

In this paper, we address the problem of MARL with vision input using reconstruction-based representation learning. We motivate the exploration of this method from a practical perspective: real-world multi-agent systems need a simple and computationally efficient method for learning the control policy. Based on this motivation, we integrate a $\beta$-VAE with MAPPO and show the strong performance of this method. We further notice that the instability of the representation can compromise the sample efficiency of this method; thus, we propose MA-TRVAE to stabilize representation learning by adding a trust-region penalty term to the objective function. This results in a simple and computationally efficient method. Experiments show that this method is more sample-efficient and stable than baselines. Furthermore, it outperforms MAPPO, utilizing state-based input, and requires only 10% more training time than MAPPO with vision input. We then scale the method to more agents and demonstrate consistent performance, whereas the other baselines exhibit degradation.

This method is a successful attempt at solving MARL with vision input using simple reconstruction-based representation learning. Due to its simplicity, computational efficiency, and scalability, we anticipate a wide range of applications for this method in multi-agent systems. However, there are some limitations of this work: (i) we evaluate the method only on one benchmark, as we are not aware of other MARL benchmarks with first-person vision input for each agent, and (ii) this work is primarily empirical, lacking the theoretical performance analysis for the method.

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

## A  EXTENDED RELATED WORK

**MARL.** MARL extends reinforcement learning to environments with multiple interacting agents, where objectives can be cooperative, competitive, or a combination of both. A straightforward strategy is to train agents independently, but such methods suffer from non-stationarity because each agent's policy updates alter the effective environment dynamics. To mitigate this issue, *centralized training with decentralized execution* (CTDE) has become the dominant paradigm. In CTDE, a centralized critic utilizes global state information during training, whereas decentralized actors rely solely on local observations during execution. Lowe et al. (2017) introduced this framework for continuous control by extending deterministic policy gradients with a centralized critic, known as the *multi-agent deep deterministic policy gradient* (MADDPG). This paradigm has since become the foundation for subsequent methods. Policy-gradient-based extensions of CTDE have proven particularly effective. Yu et al. (2022) adapted PPO to the multi-agent setting with a centralized value function, achieving stable and scalable performance in cooperative settings, known as multi-agent proximal policy optimization (MAPPO). Building on this, Kuba et al. (2021) extends trust-region methods to heterogeneous agents called heterogeneous-agent trust-region policy optimization (HATPRO) and heterogeneous-agent proximal policy optimization (HAPPO), which provides theoretical guarantees of monotonic policy improvement under CTDE and enhances sample efficiency. Value decomposition offers an alternative approach to coordination and credit assignment by factorizing a global action-value function into per-agent utilities. Rashid et al. (2018) introduce such a method called QMIX, which factorizes a global action-value function into per-agent utilities via a monotonic mixing network, enabling effective coordination in cooperative settings while maintaining scalability. More recently, transformer-based architectures have been proposed to capture temporal and inter-agent dependencies better. Wen et al. (2022) propose multi-agent transformer (MAT) that treats agents and their trajectories as sequences, applying self-attention to model interactions and dynamics. This approach offers expressive joint representations and has demonstrated enhanced performance, though at the expense of increased computational complexity. Together, these methods represent the major algorithmic families that have shaped MARL in recent years. While they have advanced the state of the art, challenges remain in scalability, sample efficiency, and robustness in high-dimensional and partially observable environments, motivating the exploration of complementary directions such as representation learning.

**Representation learning in RL and MARL.** Early approaches to representation learning for model-free RL in the single-agent case apply deep autoencoders to learn feature spaces (Lange & Riedmiller, 2010; Lange et al., 2012), but lack scalability in complex environments and require expert knowledge. Subsequent methods leverage VAEs (Kingma & Welling, 2022) for this task, where the RL agent learns policies using the latent representation as inputs (Higgins et al., 2018; Nair et al., 2018). Yarats et al. (2021) then identify that the stochasticity of VAEs damages the RL agent performance and propose to use a deterministic autoencoder. Their method, dubbed SAC+AE, was reported as achieving state-of-the-art performance at the time. Apart from reconstruction-based approaches, Laskin et al. (2020) and Stooke et al. (2021) leverage contrastive learning, but rely on designing a similarity function as well as positive and negative samples. Model-based approaches leveraging the VAE are based on world models (Ha & Schmidhuber, 2018a), which proved too complex due to several auxiliary losses (Hafner et al., 2019b)(Lee et al., 2020). In the multi-agent domain, Shang et al. (2021) combine a cross-agent attention module with an unsupervised trajectory prediction task. Song et al. (2023) introduce a masked attentive contrastive learning framework that reconstructs masked agent observations in the latent space using an attentive model and contrastive loss. This approach enables agents to leverage inter-agent correlations, leading to improvements in existing MARL algorithms. Building on this, Feng et al. (2025) use a attention-based world-model approach with a self-supervised learning objective. While these approaches show promise, they either lack generality, impose high computational demands, or introduce considerable complexity. This motivates the need for simpler methods, that are both computationally efficient and scalable, a gap that we address in this work.

**Trust region optimization in RL.** Trust region methods are a common tool used in RL to bound the size of policy updates. Kakade & Langford (2002) first introduce this idea into RL by forming a constrained optimization problem and prove a monotonic improvement guarantee for policy update. While their method is based on mixing policies, Schulman et al. (2015) introduce *trust-region policy optimization* (TRPO) that extend their method to non-linear stochastic policies, which enables the use of neural networks for solving high-dimensional control tasks. TRPO enforces the trust region

constraint by constraining the *Kullback-Leibler (KL) divergence* between the old policy and the new one. Despite its theoretical rigor, TRPO uses second-order optimization to enforce the constraint and hence scales poorly. Schulman et al. (2017a) later introduce *proximal policy optimization* (PPO) to overcome this issue by forming the trust region optimization as an unconstrained optimization problem. This allows for efficient first-order optimizations and better sample efficiency.

## B  TRUST REGION OPTIMIZATION

Trust region methods optimize an objective function by restricting the step size of updates so that the new solution does not deviate drastically from the previous one. This is typically achieved by enforcing a constraint in the optimization process or adding a penalty term to the objective function. When it comes to policy optimization, while TRPO implements the idea by solving a constrained optimization problem, which proved to be computationally costly, PPO chooses the latter approach by adding a clip penalty to the objective function as a surrogate to optimize

$$L^{clip}(\theta) = \mathbb{E}_t\left[\min\left(r_t(\theta)\hat{A}_t,\; \text{clip}(r_t(\theta), 1 \pm \tau)\,\hat{A}_t\right)\right], \tag{7}$$

where $r_t(\theta) = \frac{\pi_\theta(a_t|o_t)}{\pi_{\theta_{old}}(a_t|o_t)}$ is the ratio between the new policy $\pi_\theta(a_t|o_t)$ and the old one $\pi_{\theta_{old}}(a_t|o_t)$, and $\hat{A}_t$ is the advantage function that estimating how much the current policy is better than average. The clip operator trims the input value to keep it within the interval $[1 - \tau, 1 + \tau]$, where $\tau$ is the clip parameter. Alternatively, one can add a KL penalty as another surrogate objective

$$L^{\text{KL}}(\theta) = \mathbb{E}_t\left[r_t(\theta)\,\hat{A}_t\; -\; \beta\,D_{KL}\big(\pi_{\theta_{old}}(\cdot|o_t)\,\|\,\pi_\theta(\cdot|o_t)\big)\right], \tag{8}$$

where $D_{KL}\big(\pi_{\theta_{old}}(\cdot|o_t)\,\|\,\pi_\theta(\cdot|o_t)\big)$ is the KL divergence between the old policy and the new one, and $\beta$ is a coefficient controlling the range of the policy update.

## C  ADDITIONAL DETAILS OF THE MULTI-AGENT ENVIRONMENT

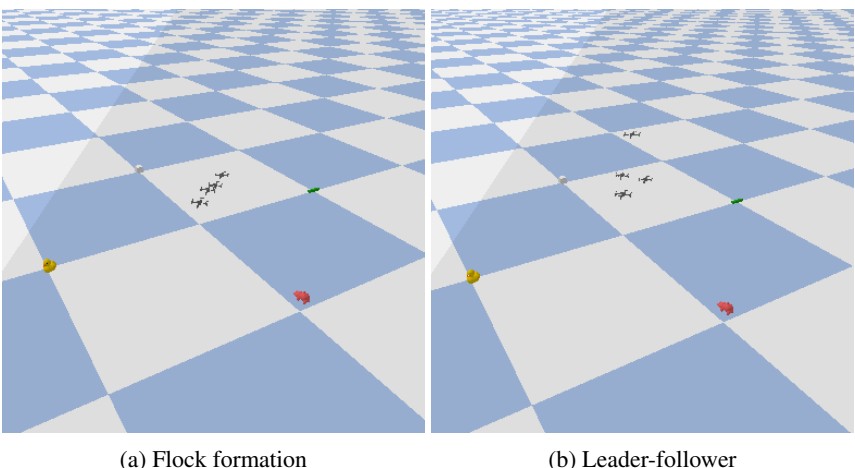

| (a) Flock formation | (b) Leader-follower |
|---|---|

Figure 7: Multi-agent tasks

In this section, we briefly describe the two multi-agent tasks, named flock and leader-follower, in MAQC. Let $\mathbf{x}_i = (x_i, y_i, z_i)$ be the position coordinates, $r_i$ be the individual reward of $i$-th agent, and $R = \sum_{i=1}^{N} r_i$ be the team reward.

- **Flock**: In the flock scenario of MAQC, the objective is for the first agent to keep its position as close as possible to a predefined location (e.g., $p$). The individual reward for the first agent is

$$r_1 = -||\boldsymbol{p} - \mathbf{x}_1||_2^2. \tag{9}$$

The individual rewards for the remaining agents are determined by their ability to track the latitude of the preceding agent. The reward is defined as

$$r_i = -(y_i - y_{i-1})^2,\tag{10}$$

for $i = 2, 3, ..., N$. That is, all drones need to follow the first drone in a line. Figure 7a shows the flock formation task captured during the MARL training.

- **Leader-follower**: In the leader-follower scenario of MAQC, the goal is to train the follower drones to track the leader drone. The leader drone is expected to keep its position as close as possible to a predefined location. The individual reward for the leader drone is

$$r_1 = -||\boldsymbol{p} - \mathbf{x}_1||_2^2.\tag{11}$$

The individual rewards for the follower drones are determined by their ability to track the position of the leader drone. The reward is defined as

$$r_i = -\frac{1}{N}||\mathbf{x}_i - \mathbf{x}_1||_2^2,\tag{12}$$

for $i = 2, 3, ..., N$. That is, all drones need to keep close to the leader drone. figure 7b shows the leader-follower task captured during the MARL training.

## D   DETAILS OF MA-TRVAE

In this section, we include a detailed algorithm of our proposed method (i.e., MA-TRVAE combined with MAPPO). The end-to-end training process of the framework is outlined in Algorithm 1, which is heavily based on MAPPO. Also, our code is mainly based on the MAPPO implementation.[2]

---

**Algorithm 1** MA-TRVAE for MAPPO

---

**Initialize** encoder $\phi$, decoder $\psi$, dummy encoder $\phi_{\text{old}}$, actor $\theta$, critic $w$, data buffer $\mathcal{D}$
**for** each episode **do**
    **for** each timestep $t = 1$ to $T$ **do**
        **for** each agent $i = 1$ to $N$ **do**
            Observe $o_t^i$, encode $z_t^i = g_\phi(o_t^i)$
            Sample action $a_t^i \sim \pi_\theta^i(a|z_t^i)$
        **end for**
        Execute joint action $a_t$, store transition in $\mathcal{D}$
    **end for**
    **for** each mini-batch from $\mathcal{D}$ **do**
        **for** each agent $i$ **do**
            Encode latent $(\mu^i, \sigma^i)$, reparametrize $z_t^i$
            Reconstruct $\hat{o}_t^i$, compute $\mathcal{L}_{\text{TR-VAE}}, \mathcal{L}_{\text{actor}}^i$
        **end for**
        Compute critic loss $\mathcal{L}_{\text{critic}}$ using global latent state $[z_t^1, ..., z_t^N]$
        Update $\theta, w$ with PPO
        Update $\phi, \psi$ with $\mathcal{L}_{\text{TR-VAE}}$
    **end for**
    Update $\phi_{\text{old}} \leftarrow \phi$
**end for**

---

## E   HYPERPARAMETERS

Table 2 contains the common hyperparameters used for this paper. Table 3 shows the hyperparameters used for each algorithm in the experiment. For training the MARL agents and conducting other experiments, we utilized a cluster equipped with an A100 GPU and 80 GB of memory.

---

[2]https://github.com/marlbenchmark/on-policy

Table 2: Common hyperparameters used for all methods.

| Hyper-parameters | Value | Hyper-parameters | Value |
|---|---|---|---|
| $\gamma$ | 0.99 | optim eps | 1e-6 |
| max grad norm | 0.5 | gain | 0.01 |
| hidden layer dim | 64 | entropy coef | 0.01 |
| use huber loss | True | rollout threads | 20 |
| episode length | 200 | batch size | 4000 |
| stacked frames | 1 | training threads | 16 |

Table 3: Hyperparameters used for MA-TRVAE, VAE, AE, MA2CL, and MAPPO.

| Algorithm / Parameters | MA-TRVAE | MA-VAE | MA-AE | MA2CL | MAPPO |
|---|---|---|---|---|---|
| critic lr | 5e-3 | 5e-3 | 5e-3 | 5e-3 | 5e-3 |
| actor lr | 5e-4 | 5e-4 | 5e-4 | 5e-4 | 5e-4 |
| ppo epochs | 5 | 5 | 5 | 5 | 5 |
| ppo clip | 0.2 | 0.2 | 0.2 | 0.2 | 0.2 |
| num mini-batch | 4 | 4 | 4 | 4 | 4 |
| num hidden layer | 2 | 2 | 2 | 2 | 2 |
| $\lambda$ | / | / | 1e-06 | / | / |
| $\beta_1$ | 1e-07 | 1e-07 | / | / | / |
| $\beta_2$ | 1e-06 | / | / | / | / |

## F    ADDITIONAL RESULTS

This section contains the results of additional experiments we conducted for this paper.

### F.1    COMPUTATIONAL EFFICIENCY

The results of the computational efficiency test, performed in both the flock formation and leader-follower tasks with the PID action setting, are shown in figure 8. We can see similar results as we saw in section 5.2.

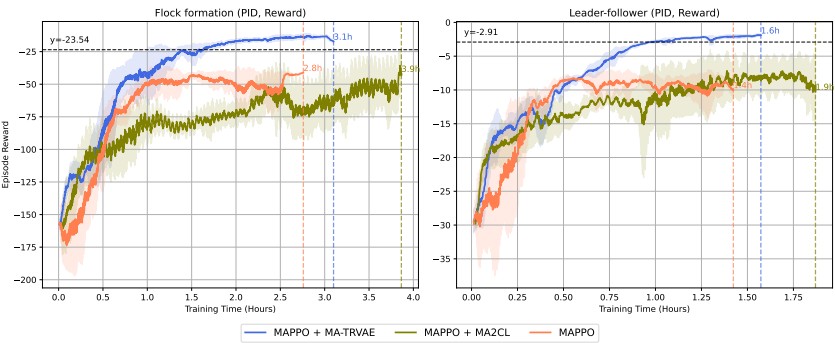

Figure 8: Episode reward vs. training time for flock formation tasks, with PID action setting. *We can see that MA-TRVAE trains faster than the MA2CL, with a better performance. The black dashed line represents the final episodic reward achieved by the MAPPO algorithm trained with proprioceptive state, which is also outperformed by MA-TRVAE.*

### F.2    SCALABILITY EXPERIMENT

The results of the scalability test performed in the leader-follower task with PID action setting are shown in figure 9. Similar to the results obtained from the flock formation experiment as shown in section 5.2, we can see that MA-TRVAE scales better when compared to other baselines.

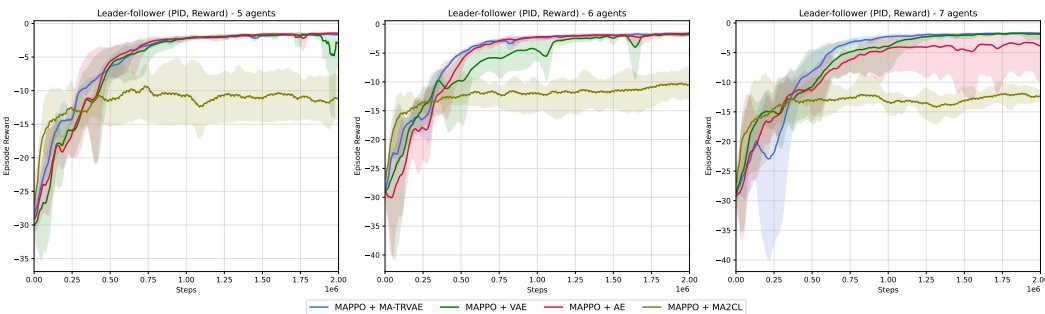

Figure 9: Episodic reward vs. training step plot for the leader-follower task with PID action setting, where the number of agents has been varied from five to seven. *We can observe that MA-TRVAE scales well with the number of agents, with a stable learning curve, outperforming the baselines*

### F.3 EXPERIMENT WITH NOISY OBSERVATIONS

In the real world, sensors mounted on physical agents are often subject to various noise sources, such as illumination changes, motion blur, occlusion, or sensor artifacts. Therefore, a robust multi-agent learning framework must not only perform well under clear observations but also retain performance when faced with degraded visual input. This experiment investigates the empirical robustness of MA-TRVAE framework, when agents are trained under additive Gaussian noisy observation conditions. This experiment is designed in two different ways, as follows, based on how the noise is added.

- **Shared noise:** A single random Gaussian noise distribution is sampled for each timestep and added uniformly to all agents' observations,

$$\tilde{\boldsymbol{o}}_t^i = \boldsymbol{o}_t^i + \boldsymbol{\xi}, \quad \boldsymbol{\xi} \sim \mathcal{N}(\boldsymbol{0}, \sigma^2 \boldsymbol{I}). \tag{13}$$

This simulates environment-level noise, such as lighting shifts or global occlusions, that affect all agents identically. The experiment is performed by taking $\sigma = 10$.

- **Agent-specific noise:** Each agent receives the random Gaussian noise independently with different standard deviations,

$$\tilde{\boldsymbol{o}}_t^i = \boldsymbol{o}_t^i + \boldsymbol{\xi}_i, \quad \boldsymbol{\xi}_i \sim \mathcal{N}(\boldsymbol{0}, \sigma_i^2 \boldsymbol{I}). \tag{14}$$

This setting simulates localized sensor noise, which will be different for the agents. The experiment is performed by taking $\sigma_1 = 10$, $\sigma_2 = 5$, $\sigma_3 = 2$, $\sigma_4 = 5$.

These noisy observations $\tilde{\boldsymbol{o}}_t^i$ are passed to the shared encoder $g_\phi$, which learns latent representations $\boldsymbol{z}_t^i = g_\phi(\tilde{\boldsymbol{o}}_t^i)$. The rest of the training pipeline remains the same. This experiment is conducted in the flock formation task with the PID action setting as mentioned in section 5.1, over two million training steps. The number of agents is set to four. In figure 10, it can be observed that, MA-TRVAE outperforms other autoencoder based methods, showing a better empirical robustness under additive Gaussian noise.

## G  PERFORMANCE OF TRUST-REGION AUTOENCODER IN SINGLE-AGENT RL

While MA-TRVAE has most potential in the multi-agent domain, it is similarly applicable in single-agent environments and provides similar benefits. To validate that proposition, we provide results from single-agent experiments. It is noteworthy that the underlying RL algorithm is different in the single-agent case than in the multi-agent case. In the latter, the algorithm is based on PPO, an on-policy algorithm, whereas in the former, the RL algorithm is soft actor-critic (SAC), an off-policy algorithm. The reason for this is that we want to evaluate the performance of TRVAE in the single-agent environment against a well-established and fair baseline (Yarats et al., 2021), which uses SAC as the base RL algorithm. Additioanlly, this can show the generality of TRVAE as it can be applied to both on-policy and off-policy RL algorithms.

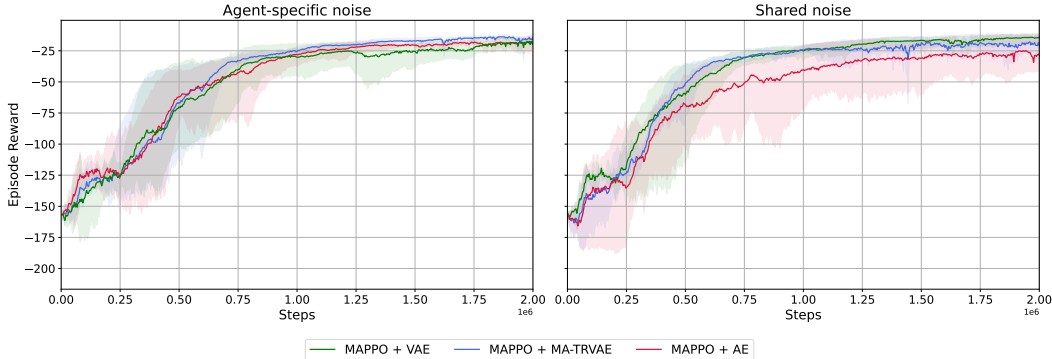

Figure 10: Episodic reward vs. training step plot for the flock formation task with PID action setting, where noise is added to the observations. *We can observe that MA-TRVAE scales well with the number of agents, with a stable learning curve, outperforming the baselines*

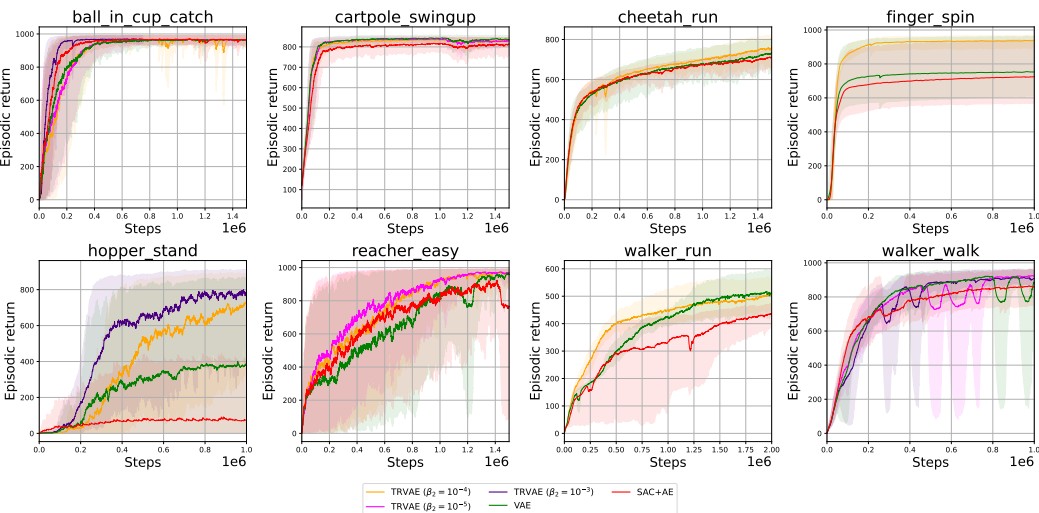

Figure 11: Episodic returns for the SAC+TRVAE compared against SAC+AE (Yarats et al., 2021) a vanilla VAE. The figure shows that the trust-region method achieves similar or better performance across all environments. Furthermore, TRVAE improves the variance of the runs.

## G.1 EXPERIMENTS AND SETUP

We evaluate the performance of our method in the single-agent domain (named SAC+TRVAE) against the performance of SAC+AE (Yarats et al., 2021) and a vanilla VAE. For the purposes of the single-agent evaluation, we consider eight different continuous control tasks from the DeepMind control suite (Tassa et al., 2018), specifically `finger_spin`, `reacher_easy`, `ball_in_cup_catch`, `cartpole_swingup`, `walker_walk`, `walker_run`, `cheetah_run`, and `hopper_stand`. We run the algorithms with the same 5 random seeds for each environment, with $\beta_1 = 10^{-7}$ for all experiments.

The setup of these experiments follows the setup and implementation of (Yarats et al., 2021), available on their GitHub page. Their repository did not include an implementation of a VAE, so we modified their code to include an implementation of a VAE and the trust-region optimization.

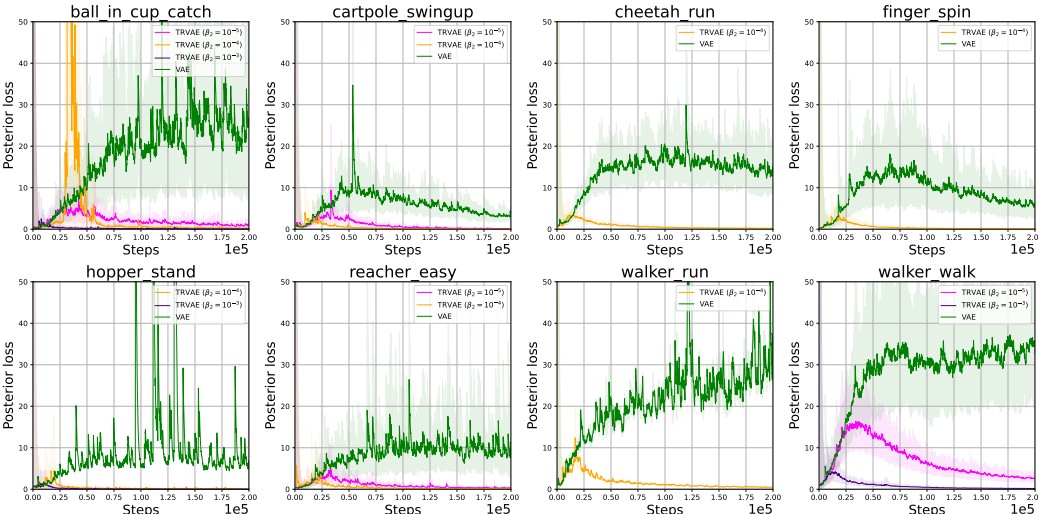

Figure 12: Posterior (trust-region) losses for TRVAE and vanilla VAE across environments. We observe that the posterior loss of the vanilla VAE is significantly higher than the trust-region method. We also see that with a higher $\beta_2$ value, the posterior loss is smaller. Smaller posterior losses imply more consistent representations between iterations.

## G.2 Results

### G.2.1 Stability and sample efficiency

The results in figure 11 illustrate both the improved performance of the RL agent with the TRVAE over both the vanilla VAE as well as SAC+AE. In `ball_in_cup_catch`, the gains in episodic returns are negligible. However, the variance is improved with the trust-region method, though a single run with $\beta_2 = 10^{-4}$ had some instability. In `cartpole_swingup`, both the vanilla VAE and SAC+TRVAE outperform SAC+AE (a result inconsistent with Yarats et al. (2021)). Gains in performance are negligible (as with `ball_in_cup_catch`), but the variance of SAC+TRVAE is better. In `cheetah_run`, all methods show similar performance, where SAC+TRVAE slightly outperforms SAC+AE and the vanilla VAE. The variance is again lowest with the trust-region. SAC+TRVAE shows the most impressive gains in `finger_spin` in terms of episodic returns and variance when compared with a vanilla VAE and SAC+AE. In one environment (`hopper_stand`), SAC+AE failed to learn effectively, whereas SAC+TRVAE shows impressive performance, as well as lower variance, compared to its vanilla counterpart. In `reacher_easy`, SAC+TRVAE out-performs other methods with several hyperparameter settings. This environment does exhibit high variance in all methods; however, the trust-region method is the most stable. In `walker_walk`, all methods showed some instability, where singular runs had bizarre dips in performance. How-ever, on average, SAC+TRVAE has the least variance with gains in performance over SAC+AE. Interestingly, the VAE also beats SAC+AE (once again inconsistent with the original paper). In `walker_run` SAC+TRVAE has the best sample efficiency, but across $2 \cdot 10^6$ steps shows simi-lar performance to the vanilla VAE (though with less variance), once more beating SAC+AE. In summary, our experiments demonstrate an increase in sample efficiency across all environments in which the algorithms were evaluated. SAC+TRVAE method exhibits the lowest variance (i.e., better stability) and improved performance in some. Interestingly, some of our findings on the performance of SAC+AE are inconsistent with those presented in the original paper.

### G.2.2 Consistency of representations

The posterior loss can be loosely utilized as a proxy for the representation drift, i.e., the higher the loss, the more the representations change between iterations. The posterior loss is the trust-region term of the TRVAE, which is to say we measure the consistency of representation by the KL divergence between the posterior distributions of successive encoders when encoding the *current*

*observations*. This measurement is done by maintaining a frozen copy of the encoder parameters from the previous step, encoding the current observations using both sets of parameters, and then measuring the KL divergence of the resulting distributions. figure 12 illustrates the posterior losses for the environments. We observe that the posterior loss of the vanilla $\beta$-VAE is the highest, as there is nothing to explicitly constrain the posterior distributions between update steps. Further, we see that the higher the $\beta_2$ value (i.e., the stronger the constraint on the latent update), the smaller the posterior loss. In `ball_in_cup_catch` we observe a spike in the posterior loss with $\beta_2 = 10^{-4}$. However, across steps, the posterior loss is smaller. The explanation for the spike is unknown, but it may be due to numerical instability or a random coincidence, as we also observe such spikes with the VAE.

It is noteworthy that a minimal posterior loss is not necessarily desirable. Too strong of a constraint on the latent update hurts the performance of the RL agent. Intuitively, the strong constraint could weaken exploration due to a lack of diversity in the representations, thereby hurting performance.

An alternative method for measuring representation drift is to compare the KL divergence in the posteriors of the current and previous autoencoders when encoding the same observation at each timestep. That method maintains invariance in the object of encoding by always encoding the same (e.g., initial) observations. Intuitively, this could provide a more accurate view of the consistency of representation by having the encoder parameters be the only change. However, empirically, both methods yield similar results.

## H  ADDITIONAL RESULTS

In this section, we provide additional results to address the reviewers' concerns. We will integrate these results into the main paper in the final version of the paper.

### H.1  RESULTS IN MULTI-AGENT MUJOCO

In this section, we show experiment results for Multi-agent Mujoco with state-based observation. We compare MA-TRVAE against MAPPO and MAPPO + MA2CL with the hyperparameters found in the MA2CL paper (Song et al., 2023). Due to the limited time, we use the same $\beta_1$ and $\beta_2$ values we used in the MAQC environment for MA-TRVAE and keep the other MAPPO-related hyperparameters identical to the MAPPO baseline. We emphasize that there are most likely better $\beta_2$ values for MA-TRVAE with state-based observations.

We test MA-TRVAE in five Multi-agent Mujoco domains: 3x1 Hopper, 6x1 Half_cheetah, 6x1 Walker, 8x1 Ant and 10x2 Swimmer. We show results in Fig. 13. Even though MA-TRVAE is not designed for state-based observations, it outperforms MAPPO and MAPPO + MA2CL by a large margin in higher-dimensional domains (6x1 Half_cheetah, 6x1 Walker, 8x1 Ant and 10x2 Swimmer). While it shows slightly worse performance on the lower-dimensional task (3x1 Hopper), we observe lower training variance. This shows that reconstruction-based representation learning is an effective method for state-based MARL.

### H.2  RESULTS IN GOOGLE RESEARCH FOOTBALL

In this section, we show results for the Google Research Football (GRF) benchmark (Kurach et al., 2020) with visual observation. In this benchmark, each agent receives RGB frames that contain global information about the environment and controls a player with a high-level action space (moving in different directions, passing, shooting, etc.). Since the observation is not local, it does not fit the decentralized POMDP setting. However, this is the closest setting we could find to test our method. We compare MA-TRVAE with MAPPO and MAPPO + MA2CL.

We observe from Fig. 14 that MA-TRVAE outperforms MAPPO after 2.5 M steps and converges to a similar reward as MA2CL. We did not observe less variance from MA-TRVAE. This is likely caused by the lack of hyperparameter search for TRVAE in this environment. Due to limited time and computing resources, we prioritize fine-tuning the hyperparameters of the underlying MAPPO rather than those of TRVAE, since the baseline MAPPO does not learn with the hyperparameters from the MAQC benchmark.

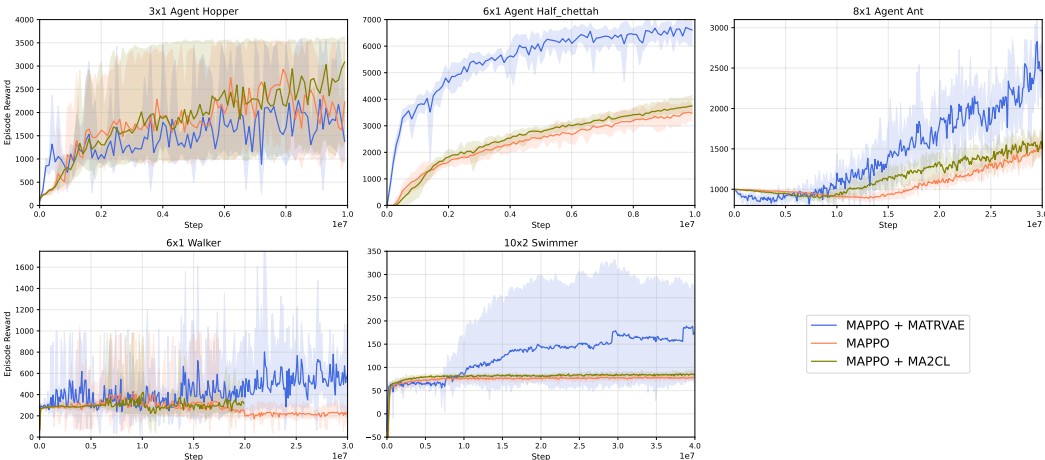

Figure 13: Results on the state-based multi-agent Mujoco. *Experiments are conducted in the hopper, half-cheetah, walker, swimmer and ant domains. MA-TRVAE outperforms the baselines by a large margin in higher-dimensional tasks. The result for MA2CL after 2M in walker domain is missing since we run out our of computing resource.*

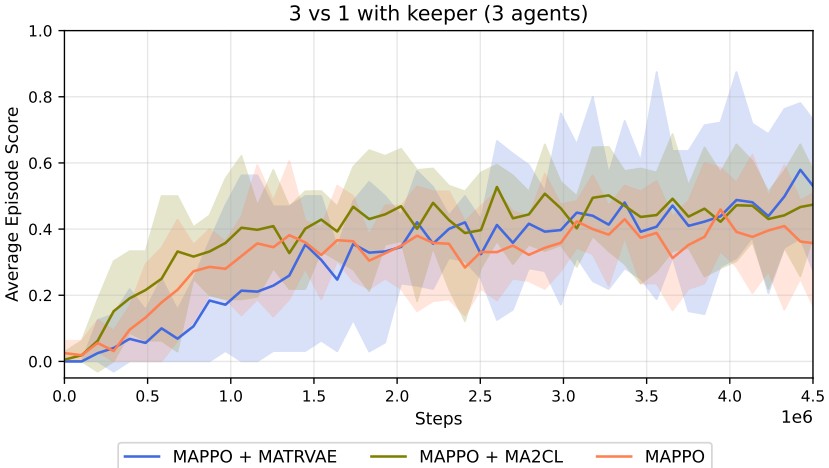

Figure 14: Results on the GRF benchmark.

The key changed hyperparameters are the number of PPO epochs and the number of mini-batches. This results in 30 updates to the actor, critic, and VAE parameters per rollout. In MAQC, this number is 20. However, we use the same hyperparameters from MAQC for TRVAE, which likely causes a suboptimal representation learning in GRF. Additionally, as stated before, there is a large difference in the observations between the two environments, so we expect that the same hyperparameters for TRVAE would not lead to good performance. This can be observed from the loss curve of MA-TRVAE (see Fig. 15). We found that the total TRVAE loss and reconstruction loss drop much faster than the curve shown in the MAQC environment (see Fig. 16), suggesting a high learning rate for TRVAE. Also, we observe that the trust-region loss is larger than that in MAQC. This suggests we need a larger $\beta_2$.

Lastly, we want to stress that the observation in GRF makes the representation learning in this benchmark similar to the single-agent tasks we showed in Section G, where the observation contains global information and does not change from one agent to another. This potentially mitigate the benefit of having TRVAE, since the observation is less diverse and more static. We can understand this from how the pixels changes from one step to another. Since the observation for GRF is the screenshot of the whole scree, these changed pixels are positions of players and balls, which only covers a small

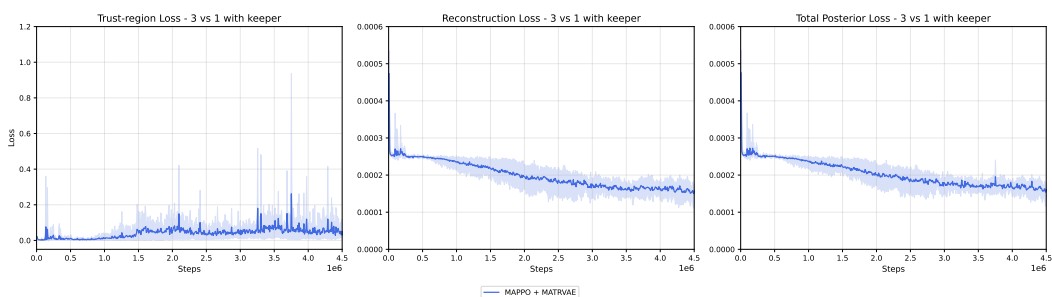

Figure 15: Results on the GRF benchmark showing the losses of MA-TRVAE.

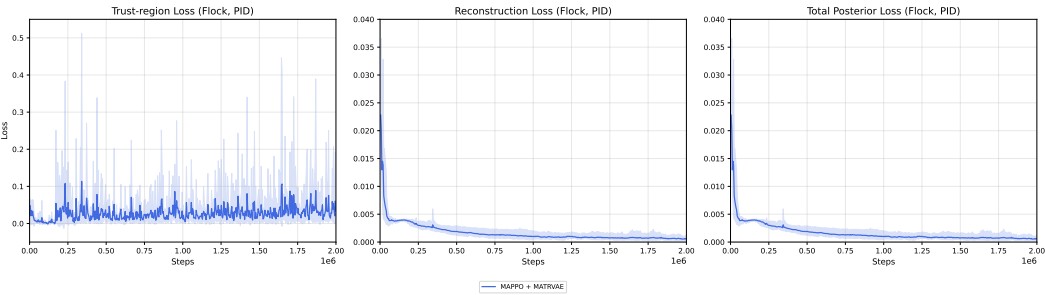

Figure 16: Results on the flock task with PID action setting showing the losses of MA-TRVAE.

fraction of total pixels. And because every agent share the same global observation, the input to the TRVAE is multiple identical frames, which likely results in an overfitting for reconstruction (this can explain why it has a extremely low reconstruction error in Fig. 15 ) instead of learning a good representation.

## H.3 ABLATION STUDY

In this section, we show the results of an ablation study on $\beta_2$ to show the effect of the trust-region constraint strength. We conduct experiments on the flock formation task with a PID action setting. We vary $\beta_2$ from $1 \times 10^{-1}$ to $1 \times 10^{-8}$ and keep the other hyperparameters the same.

Fig. 17 shows the performance of MA-TRVAE with different $\beta_2$ values. It is clearly visible that a large $\beta_2$ leads to more stable training but poorer performance, while a smaller $\beta_2$ leads to less stable training but better performance. This is attributed to either under- or over-constrained representations. While a large $\beta_2$ makes the representation more stable, it leads to poor exploration in the latent space, as the representation distribution is forced to remain close to the previous one. On the other hand, a small $\beta_2$ allows more exploration in the latent space and tends to show better performance for MARL. We do not include the performance of $\beta_2 = 1e - 1$, since it does not show any sign of learning.

## H.4 COMPARISON TO MAT

In this section, we compare our method to MAT (Wen et al., 2022) as MAPPO + MA2CL performs significantly worse than reported, and MAT could be a more reliable SOTA method to compare against. To show the full comparison, we include MAPPO + MA2CL. Nevertheless, we would emphasize this comparison might not be fair, since MAT uses a high-capacity transformer for the encoder, while MA-TRVAE uses a small CNN.

In Fig. 18, we show results of MAT in 2 MAQC tasks with 3 different action settings (see section 5.1 lines 343 to 346 for the difference between action settings). We observe that MA-TRVAE performs better than MAT in both flock formation and leader-follower task with PID action mode, while slightly worse than MAT with the other two lower-level action settings. In the flock formation task with DYN action setting, MAT additionally converges to a higher reward than MA-TRVAE.

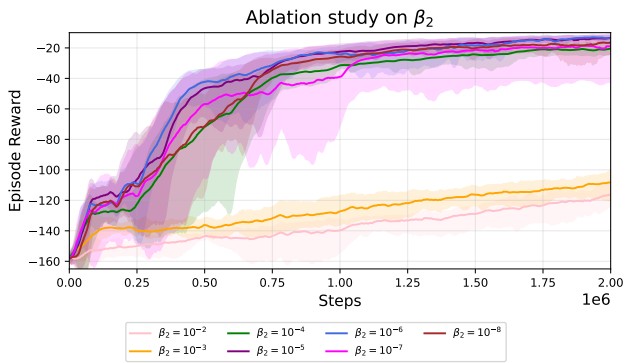

Figure 17: Ablation study for $\beta_2$ values. *Large $\beta_2$ values lead to more stable yet poor performance, while small ones tend to be more unstable but perform better.*

However, we observe MA-TRVAE consistently shows a lower variance than MAT, thanks to the stable representation learning.

We next conduct a scaling-up experiment for MAT in the flock formation task with PID action setting. The results are shown in Fig. 19. Similar to MAPPO + MA2CL, we observe that MAT's performance degrades as the number of controlled agents increases, and it shows the largest variance among all methods. While MA-TRVAE maintains a rather stable performance when scaling up the number of agents.

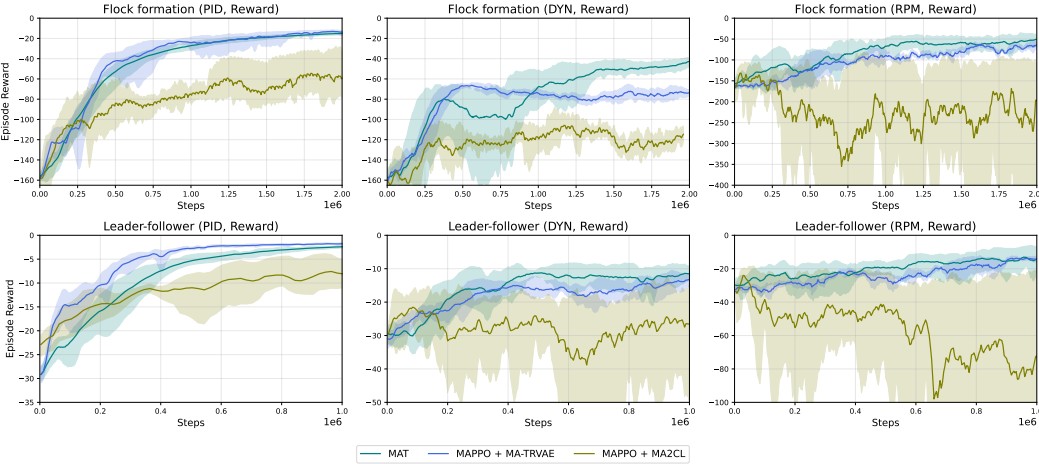

Figure 18: Comparison to MAT in MAQC tasks. *MA-TRVAE performs better than MAT in both MAQC tasks with PID action setting. For the other action settings, MA-TRVAE shows a slightly worse performance than MAT except flock formation with DYN action setting. In addition, MA-TRVAE shows a consistently lower variance in all experiments, thanks to the stable representation learning.*

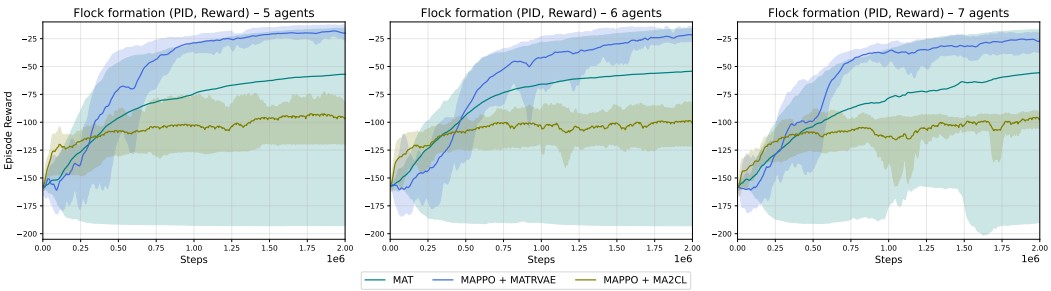

Figure 19: Scaling-up comparison to MAT. *MAT shows a performance drop and the largest variance among all methods, when scaling up the number of agents. On the contrary, MA-TRVAE maintains a similar performance and the lowest training variance.*