# OpenReview forum: "Efficient and scalable MARL from images by trust-region autoencoders"
_ICLR.cc/2026/Conference — Submitted to ICLR 2026_

### Official Review · Reviewer_me5b · 2025-10-29

**Soundness:** 3
**Presentation:** 3
**Contribution:** 2
**Rating:** 6
**Confidence:** 3

**Summary:**

This paper proposes MA-TRVAE, a Multi-agent Trust Region Variational Autoencoder for vision-based multi-agent reinforcement learning (MARL). The core motivation is that reconstruction-based representation learning (e.g., VAE) is simple and efficient but unstable when jointly trained with policy updates. The authors observe that smaller $\beta$ in $\beta$-VAE improves task performance but leads to unstable latent updates, harming policy learning. To mitigate this, they introduce a trust region penalty on latent posterior updates, constraining the representation change between consecutive steps. The resulting algorithm improves stability, sample efficiency, and scalability compared to MAPPO.

**Strengths:**

1. **Simple yet elegant method.**
   Adding a KL penalty between consecutive posteriors is a minimal but effective extension that is computationally light.

2. **Empirical insights.**
   The analysis of $\beta$’s influence on stability and performance is detailed and well-motivated, providing clear intuition behind the necessity of the trust-region regularization.

3. **Clarity.**
   The paper is clearly written and easy to follow, especially in methodology and experimental setup.

**Weaknesses:**

1. **Technical novelty is moderate.**
   While the trust-region penalty for latent stability is reasonable, it is a straightforward adaptation of existing concepts (TRPO/PPO) to the representation space. The core mechanism resembles regularized VAE updates.

2. **Limited evaluation scope.**
   Experiments are conducted solely on the MAQC benchmark. Although this is a strong testbed, additional validation on diverse MARL domains (e.g., SMAC with vision input) would strengthen generality.

3. **Lack of theoretical justification.**
   The method is empirically effective but lacks formal analysis (e.g., convergence or stability bounds for latent updates).

**Questions:**

1. Could the authors further analyze the sensitivity of $\beta_2$? It might help understand how strong the constraint should be for balancing stability vs. expressivity.

2. Is the same trust-region idea applicable to contrastive or predictive representation learners (e.g., MA2CL)? Some comparison or discussion would be useful.

---

> ### Author Response · Authors · 2025-11-21
>
> We thank the reviewer for the positive evaluation of our work. Below we mainly address weakness 2 and the proposed questions.
>
> **W2**
>
> We acknowledge the limited scope of the experimental evaluation and are working on extending it to more environments. We have already collected results from three multi-agent Mujoco domains and are conducting an experiment in Google Research Football, a visual-based MARL environment. The preliminary results in the multi-agent Mujoco environment show that MA-TRVAE largely outperforms baselines in two domains (6x1 Half_cheetah and 8x1 Ant) and demonstrates better stability in another domain (3x1 Hopper). For SMAC with vision input, we didn't find too much information about it online, and have the problem of turning the visual observation on. However, we would like to try it if the Google Research Football experiment fails. We will upload the results once extra experiments are completed.
>
> **Could the authors further analyze the sensitivity of
> $\beta_2$? It might help understand how strong the constraint should be for balancing stability vs. expressivity.**
>
> We conducted an ablation study on $\beta_2$ value from 1e-1 to 1e-8 and found that the MARL agent's performance was indeed affected by the strength of the trust-region constraint. While a large $\beta_2$ makes the representation more stable, it leads to poor exploration in the latent space, as the representation distribution is forced to remain close to the previous one. On the other hand, a small $\beta_2$ allows more exploration in the latent space and tends to show better performance for MARL. We will upload the ablation study result next week.
>
> **Is the same trust-region idea applicable to contrastive or predictive representation learners (e.g., MA2CL)? Some comparison or discussion would be useful.**
>
> Conceptually, the trust-region constraint would also work for contrastive and predictive representation learning if there is no strong regulation in the output space. This can be understood from the vanilla VAE, where the coefficient of the KL term is 1. When there is a strong constraint in the latent space, the representations do not drift too much between steps, since both representations are pushed toward the same latent space. Our trust-region idea works for the $\beta$ VAE because we observe that the strong constraint in the latent space harms MARL agent performance, so we choose a weaker constraint, leading to representation drift between steps.

---

> ### Comment · Reviewer_me5b · 2025-11-22
> **Thanks for the response**
>
> Thanks for the authors’ response! Regarding SMAC, I am referring to the StarCraft Multi-Agent Challenge benchmark. The Google Research Football experiment with visual input is also acceptable to me. Since the results for these high-dimensional experiments, as well as the sensitivity analysis for $\beta_2$, are not yet available, I will maintain my current score.
>
> **Reference:**
>
> [1] Samvelyan, Mikayel, et al. "The starcraft multi-agent challenge." arXiv preprint arXiv:1902.04043 (2019).

---

> ### Author Response · Authors · 2025-12-03
>
> We thank the reviewer for their positive response and the reference for SMAC. We found there is a way to use pixel observation in SMAC, but it only gives the global observation of the screen and requires manual cropping to get local observation for agents. We therefore opted for the Google Research Football environment, which provides the same global observation for each agent.
>
> We have updated our paper with additional results in section H.

---

### Official Review · Reviewer_cW5t · 2025-10-30

**Soundness:** 3
**Presentation:** 3
**Contribution:** 3
**Rating:** 4
**Confidence:** 3

**Summary:**

This paper introduces MA-TRVAE (Multi-Agent Trust-Region Variational Autoencoder), a new method to improve sample efficiency and stability in vision-based multi-agent reinforcement learning (MARL). The method incorporates a trust-region constraint into the VAE objective, ensuring smooth updates of latent representations. Combined with MAPPO, the method achieves better performance across multi-agent control tasks such as flock formation and leader-follower control

**Strengths:**

The paper avoids complex attention or contrastive mechanisms while still outperforming prior visual MARL baselines. Its adaptation of trust-region optimization to the representation learning stage is novel and well-motivated, offering a new perspective on stabilizing learned visual embeddings.

**Weaknesses:**

* The main limitation is the narrow empirical scope: all experiments are conducted in the MAQC drone environment, leaving uncertainty about generalization to other multi-agent domains (e.g., manipulation, different embodiments).
* Moreover, although the method is computationally efficient, hyperparameter sensitivity of β₁ and β₂ is not deeply analyzed—these may be crucial for balancing reconstruction quality and stability.

**Questions:**

* How sensitive is MA-TRVAE to the trust-region coefficient β₂? Could different values lead to either under- or over-constrained representations?
* Since all tasks are vision-based drone control, have the authors tried other domains (e.g., multi-agent MuJoCo or social navigation) to confirm scalability?
* How does the trust-region constraint compare against simpler stabilization methods such as target encoders or temporal smoothing of latent features?

---

> ### Author Response · Authors · 2025-11-21
>
> We thank the reviewer for the evaluation of our work.
>
> **W1**
>
> While we acknowledge the limited scope of our experimental evaluation, there are unfortunately not many visual MARL environments with decentralized observations. Nevertheless, to address this concern, we have extended our evaluation to the state-based Multi-agent MuJoCo benchmark and are running experiments in Google Research Football. Preliminary results from three Multi-agent Mujoco domains show that our method can indeed generalize across domains (see reply to Q2 for more details). We will share the results from Multi-agent MuJoCo and Google Research Football once the training is done.
>
> **W2**
>
> We agree that the effect of $\beta_1$ and $\beta_2$ is crucial. We therefore did an ablation study for $\beta_2$ values from 1e-1 to 1e-8. A large $\beta_2$ value leads to a poor agent performance, while weak ones tend to have a good performance. For the ablation study of $\beta_1$, we have shown it in Fig.2, and the effect of the $\beta_1$ value is analyzed in section 3.1.
>
> **How sensitive is MA-TRVAE to the trust-region coefficient $\beta_2$? Could different values lead to either under- or over-constrained representations?**
>
> From the ablation study we conducted, we can see that different values indeed affect MARL agent performance significantly, and this is attributed to either under- or over-constrained representations. While a large $\beta_2$ makes the representation more stable, it leads to poor exploration in the latent space, as the representation distribution is forced to remain close to the previous one. On the other hand, a small $\beta_2$ allows more exploration in the latent space and tends to show better performance for MARL. The result of the ablation study for $\beta_2$ will be uploaded next week.
>
> **Since all tasks are vision-based drone control, have the authors tried other domains (e.g., multi-agent MuJoCo or social navigation) to confirm scalability?**
>
> Yes, we have tried multi-agent Mujoco. The preliminary results across three Mujoco domains show the method's generality even in a state-based environment. In particular, we observe that we can significantly outperform MA2CL and MAPPO on two high-dimensional benchmarks (6x1 Half_cheetah and 8x1 Ant) and perform slightly worse than MAPPO in a lower-dimensional benchmark (3x1 Hopper), with lower variance than the baselines.
>
> **How does the trust-region constraint compare against simpler stabilization methods such as target encoders or temporal smoothing of latent features?**
>
> We appreciate the reviewer raising the question. We did not compare empirically with them, but we can provide a conceptual comparison with these general stabilization methods and argue that the trust-region addresses specific challenges in Visual MARL that these simpler methods do not.
>
> While target encoders (e.g., EMA) stabilize the network weights, in deep networks, small weight perturbations can still lead to large shifts in the output distribution, i.e., the stability in parameter space does not imply stability in function space. Our method acts directly in the function space, explicitly constraining the semantic shift of the latent representation. This guarantees that the input space of the policy does not drift largely, which is critical for MARL agents to learn.
>
> Temporal smoothing enforces representation consistency between time steps ($t$ and $t-1$), which assumes the state does not change too much. However, this is a detrimental assumption for visual-based MARL, where state transitions can be abrupt (e.g., sudden agent interaction). Our method instead enforces representation consistency between updates (current encoder vs. old encoder) on the *same* observation. This allows the agent to model fast, complex dynamics without losing information about rapid environmental changes.

---

### Official Review · Reviewer_16Nd · 2025-10-31

**Soundness:** 2
**Presentation:** 2
**Contribution:** 2
**Rating:** 4
**Confidence:** 3

**Summary:**

This paper introduces MA-TRVAE, a trust-region-regularized variational autoencoder for vision-based multi-agent reinforcement learning (MARL). The method targets the issue that, while autoencoder-based  state representations are simple and efficient, they suffer from instability and sample inefficiency in MARL due to representation drift. MA-TRVAE applies a KL-divergence-based trust region constraint to stabilize latent representations between optimization steps, enabling improved learning efficiency and scalability. Results on multi-agent quadcopter control (MAQC) tasks demonstrate improved sample efficiency, stability, scalability, and computational efficiency relative to multiple baselines, including MAPPO, MA2CL, and vanilla (beta-)VAE approaches.

**Strengths:**

- **Simplicity and Clarity of Approach**: The integration of a trust-region KL penalty into a VAE framework for MARL is conceptually simple yet effective, elegantly combining ideas from trust-region policy optimization and representation learning.
- **Analysis of Representation Drift**: Identifying and quantifying representation instability (via posterior KL divergence) as a key performance bottleneck in MARL is a valuable contribution. **Figure 2** and **Figure 4** clearly demonstrate how the trust-region constraint stabilizes representation drift in practice, reinforcing the core motivation behind MA-TRVAE.
- **Computational Efficiency**: Analyses of wall-clock time and training speed (e.g., **Figure 1** and **Figure 8**) show that MA-TRVAE is not only more sample-efficient but also significantly less resource-intensive than its strongest baseline (MA2CL).
- **Reproducibility**: The paper provides thorough implementation details, algorithmic pseudocode, and complete hyperparameter tables, along with a link to the source code, enhancing transparency and reproducibility.

**Weaknesses:**

1. **Single Benchmark and Limited Environment Diversity**: The main experiments rely solely on the multi-agent quadcopter control (MAQC) simulator, focusing primarily on flocking and leader-follower tasks (see Section 5 and **Appendix C, Figure 7**). This considerably limits the claims of generality, applicability, and robustness to other vision-based MARL scenarios.
2. **Over-reliance on Reconstruction-based Baselines**: While strong baselines (MAPPO variants and MA2CL) are included, several recent state-of-the-art vision-based MARL methods (e.g., those utilizing transformers, cross-agent representation learning, or newer contrastive regularization techniques) are not compared. The absence of these may weaken the empirical claims. At minimum, it would be useful to compare against recent MAE-based approaches such as [1] or [2], as well as alternative MARL algorithms like [3] or [4].
3. **Limited Discussion of Failure Cases and Robustness**: Although **Appendix F.3, Figure 10** demonstrates that MA-TRVAE is robust to observation noise, there is no explicit analysis of more challenging conditions such as adversarial noise, partial observability, or real-world sensor failures.
4. **Ablation Studies**: The paper does not include ablations for different encoder/decoder architectures, latent dimensions, or the removal of individual regularization terms, making it difficult to fully assess the contribution of each component of the proposed method.


[1] **Kang, S., Lee, Y., Kim, G., Chong, S., & Yun, S.-Y. (2025).** MA²E: Addressing Partial Observability in Multi-Agent Reinforcement Learning with Masked Auto-Encoder. In Proceedings of the 13th International Conference on Learning Representations (ICLR 2025). [https://openreview.net/forum?id=klpdEThT8q](https://openreview.net/forum?id=klpdEThT8q)



[2] **Feng, J., Chen, M., Pu, Z., Xu, Y., & Liang, Y. (2025).** MA2RL: Masked Autoencoders for Generalizable Multi-Agent Reinforcement Learning. arXiv preprint arXiv:2502.17046.
[https://arxiv.org/abs/2502.17046](https://arxiv.org/abs/2502.17046)

[3] **Wen, M., Kuba, J. G., Lin, R., Zhang, W., Wen, Y., Wang, J., & Yang, Y. (2022).** Multi-agent reinforcement learning is a sequence modeling problem. In Proceedings of the 36th International Conference on Neural Information Processing Systems (NeurIPS 2022), 1201. Curran Associates Inc., Red Hook, NY, USA.



[4] **Guo, X., Shi, D., & Fan, W. (2023).** Scalable Communication for Multi-Agent Reinforcement Learning via Transformer-Based Email Mechanism. In Proceedings of the 32nd International Joint Conference on Artificial Intelligence (IJCAI-23), 126–134. International Joint Conferences on Artificial Intelligence Organization.

**Questions:**

1. Were alternative trust-region penalty formulations (e.g., symmetric KL, Wasserstein distance, or JS divergence) explored, and how did they perform in preliminary experiments?
2. Is the scalability improvement shown in **Figure 6** and **Figure 9** consistent when scaling to a larger number of agents (e.g., 10, 20, or more)? Do training times remain reasonable under such conditions?
3. Regarding robustness to noise (**Appendix F.3**, **Figure 10**), what happens when both shared and independent noise sources are combined, or when the noise statistics are non-Gaussian or time-varying?

---

> ### Author Response · Authors · 2025-11-21
> **Reply to weaknesses**
>
> We thank the reviewer for the comprehensive review of our work and suggestions for strengthening the paper.
>
> **Single Benchmark and Limited Environment Diversity**
>
> We acknowledge the limited scope of the environments. Nevertheless, we would like to emphasize the difficulty of finding a visual-based MARL benchmark, as reviewer Enue pointed out. We understand that the limited diversity would weaken the generality of our method; therefore, we tested our method on the Multi-agent Mujoco benchmark and are running experiments on the Google Research Football benchmark, which is a visual-based MARL environment.
>
> We have tested our algorithm in three Multi-agent Mujoco domains. We find that MA-TRVAE outperforms both MA2CL and MAPPO in two domains (8x1 Ant and 6x1 Half_cheetah) by a significant margin. In the 3x1 hopper, it shows a slightly worse performance than the baselines, but shows lower variance. Thus, also in the state-based setting, we see the potential of our method, particularly in high-dimensional spaces. We will upload results together with those for Google Research Football in the next week.
>
> **Over-reliance on Reconstruction-based Baselines**
>
> We thank the reviewer for proposing alternative baselines for comparison. We therefore ran experiments with MAT and compared our method to it. The preliminary result shows MAT is a stronger baseline than MA2CL. Our method achieves similar results of MAT in PID control mode and slightly worse in the other two action modes in both quadcopter tasks. Nevertheless,   we observe MA-TRVAE shows lower variance and less training time compared to MAT. In addition, we found MAT scales poorly in terms of the number of agents with worse performance and larger variance than MA-TRVAE. We will upload the results of MAT next week.
>
> For the proposed MAE-based methods, we would argue that they are not a good baseline to compare with, as they address a different challenge than ours. While both MAE-based methods learn a representation of the observation (state-based), they tackle the problem of partial observability rather than learning a good representation from images. Both methods use MAEs to learn a representation that can infer the global state from the local observation in a state-based MARL setting. It is not trivial to apply these two methods in a visual-based MARL setting. Nevertheless, we believe applying MAE in visual-based MARL is a promising research direction.
>
> **Limited Discussion of Failure Cases and Robustness**
>
> We thank the reviewer for proposing multiple more challenging conditions to test the robustness of MA-TRVAE. However, this work focuses more on sample efficiency and computational efficiency of visual-based MARL. And we would argue that the ego-centric view of agents is already partially observable.
>
> **Ablation Studies**
>
> It is true that we did not explicitly include ablation studies in the paper. Nevertheless, we show a comparison of the vanilla $\beta$ VAE with different $\beta$ values in Figure 2 to check the effect of the $\beta$ value. We then compare our MA-TRVAE with vanilla $\beta$ VAE, which can be seen as an MA-TRVAE with $\beta_2 = 0$. Also, we compare with the deterministic autoencoder proposed by Yarats et al. [1]  to show the effect of having variational autoencoders in the MARL setting.
>
> Similar to MAPPO and MA2CL, our method is agnostic to the model architecture and latent dimension. Therefore, we believe it is not necessary to do an ablation study on them.
>
> Our core contribution is the trust-region constraint in Eq.6; we thus recognize that testing different $\beta_2$ values is important to assess the effect of varying trust-region constraint strengths. We conducted ablation study for $\beta_2$ value from 1e-1 to 1e-8, and we can see that different values indeed affect MARL agent performance significantly. While a large $\beta_2$ makes the representation more stable, it leads to poor exploration in the latent space, as the representation distribution is forced to remain close to the previous one. On the other hand, a small $\beta_2$ allows more exploration in the latent space and tends to show better performance for MARL. We will upload the result of the ablation study next week.
>
> [1] Yarats, Denis, et al. "Improving sample efficiency in model-free reinforcement learning from images." Proceedings of the aaai conference on artificial intelligence. Vol. 35. No. 12. 2021. https://arxiv.org/abs/1910.01741

---

> ### Author Response · Authors · 2025-11-21
> **Reply to questions**
>
> **Were alternative trust-region penalty formulations (e.g., symmetric KL, Wasserstein distance, or JS divergence) explored, and how did they perform in preliminary experiments?**
>
> We did not explore other trust-region penalty formulations because the KL divergence is the most natural and consistent choice for our setting. In vanilla $\beta$-VAE training, the latent distribution is already regularized toward a standard Gaussian via a KL penalty. Introducing a different distance measure for the trust-region constraint would create a mismatch between the objectives used for representation learning and stabilization, potentially complicating optimization and leading to conflicting gradients. For this reason, we chose to use the KL divergence consistently across both penalty terms in the loss function.
>
> **Is the scalability improvement shown in Figure 6 and Figure 9 consistent when scaling to a larger number of agents (e.g., 10, 20, or more)? Do training times remain reasonable under such conditions**
>
> We did not try to scale beyond 7 agents because training takes a long time. When it comes to more agents, the bottleneck in training time is not MARL training but the simulation speed.
>
> **Regarding robustness to noise (Appendix F.3, Figure 10), what happens when both shared and independent noise sources are combined, or when the noise statistics are non-Gaussian or time-varying?**
>
> That's an interesting question, and we would like to explore it if we have more time. But for now, we would still emphasize that the focus of the work is efficiency in MARL.

---

> > ### Comment · Reviewer_16Nd · 2025-11-26
> >
> > Thank you for your response! You have addressed all of my major concerns. In good faith, I will increase my score to 6, since the experiments are not yet available.
> >
> > Regarding robustness to noise: while the main contribution of the paper focuses on policy quality and sample efficiency, Appendix F3 also claims that the model is robust to observation noise. I find this claim misleading because Gaussian noise is a convenient mathematical approximation but does not generally reflect the complexity of real-world noise. Robustness to noise is typically supported either by theoretical guarantees or by empirical evidence using more complex and realistic noise models (and even then, it may only demonstrate empirical robustness rather than true robustness).
> >
> > I believe it would strengthen the paper to revise this claim to something more precise, such as “empirical robustness under additive Gaussian noise.”

---

> > > ### Author Response · Authors · 2025-12-03
> > >
> > > We thank the reviewer for your response and the increase of the score! Please check out our new results in the updated paper (see section H).
> > >
> > > Regarding robustness to noise, we agree that our claim is too strong and have revised it in the paper (see lines 884 and 905). We thank the reviewer for helping us make our paper more precise.

---

### Official Review · Reviewer_Enue · 2025-11-01

**Soundness:** 2
**Presentation:** 2
**Contribution:** 2
**Rating:** 2
**Confidence:** 4

**Summary:**

The paper investigates representation learning for MARL with autoencoders. Section 2 sets up the preliminaries and background, on the RL objective in eq (1) for a decentralized POMDP. Then eq (2) summarizes multi-agent PPO to optimize the policies for that objective, which also estimates a value function with eq (3). Then, eq (4) highlights the \beta-VAE objective for learning a latent representation. Section 3 starts from an extension of the single-agent setting in Yarats et al. using a \beta-VAE for representation learning, finding two related trends 1) larger gaps between different \beta values than in the single-agent setting, 2) less training stability from stochasticity in the representations, and 3) other instabilities of the changing representations. To address these, section 4 proposes a trust-region around the VAE updates so it does not move by too much, which is integrated via a penalty term in eq (6), controlled by \beta_2. The experiments investigate two settings on multi-agent quadcopter control from pixels.

**Strengths:**

I generally like the idea of representation learning for RL, and I think using a \beta-VAE is a reasonable way of obtaining representations. This paper proposes a natural extension from the single-agent setting of Yarats et al., and provides clear experiments of the performance in the multi-agent settings for quadcopter control. Conceptually it is appealing and straightforward.

**Weaknesses:**

While improved representations for MARL is appealing, I have some concerns with the experimental settings that hold me back from accepting the paper. This is because I feel a strong experiment section is necessary for this kind of paper, as conceptually the idea of \beta-VAE representations and a trust-region stabilization is not sufficient.

Firstly is that the quadcopter settings are limited.
The conclusion and limitations section of the paper acknowledges this, and I would like to acknowledge the difficulty of finding an isolated decentralized multi-agent visual RL environment, since this is often a small piece in a much larger experimental system that is not cleanly isolated as a benchmark.
However, there are some other visual multi-agent RL settings that are great demonstrations and more connected to experimental robotics, such as [MAANS](https://sites.google.com/view/maans) and [robot soccer](https://arxiv.org/abs/2405.02425).
I would find experimental comparisons in settings such as these significantly more convincing, although still acknowledge the difficulty of not having clear gym-like environments for these settings.

Finally, scoping to only the quadcopter experiments, there aren't many direct comparisons to the previously-published results in MA2CL --- the MA2CL results reported in the paper under submission seem significantly worse than reported. Furthermore, only the flock and leader-follower environments from MA2CL are used, and not the state-based multi-agent starcraft and mujoco environments. The paper under submission argues they want to focus on visual settings, but the method would still work will in the state-based settings MA2CL evaluate in, and this seems scientifically interested to understand how the \beta-VAE compares to contrastive learning. I would find it significantly more convincing to have more direct comparisons to MA2CL.

**Questions:**

I am very open to further discussing my review throughout the discussion period, especially the question of if the current set of experiments.

And out of curiosity, I do have one other question on the alternative to the trust-region constraint: did you also consider using an exponential moving average (EMA) on the weights of the VAE components to help prevent it from changing too much?

---

> ### Author Response · Authors · 2025-11-21
>
> We thank the reviewer for the constructive feedback and encouraging review.
>
> We acknowledge the limited scope of the experimental setting and thank the reviewer for proposing potential environments to test our method.
> Indeed, additional demonstrations in multi-agent robotic tasks would be more convincing than evaluations on a single quadcopter control environment. While the reviewer’s suggested paper, MAANS and robot soccer, show strong benchmarks for multi-agent systems, several practical constraints currently prevent us from evaluating our method in these settings.
>
> - **MAANS** uses a setting for multi-agent visual exploration, which typically requires planning and is not trivial for end-to-end MARL methods. In fact, we find that all baselines in the paper are planning-based methods, including the proposed MAANS itself. While we could integrate our representation module into MAANS’s multi-stage pipeline, this would make it only a small component of a larger planning system, preventing a clear evaluation of our method itself. Also, it is not trivial to figure out how to do the integration, given the short amount of time.
> - **Robot soccer** provides a simulation environment for robot football games. The environment provides an egocentric view for each agent and allows direct control of the robot joints. This is an ideal testbed for our method, but, to our knowledge, the authors did not open-source the training environment. This makes the evaluation in such an environment very challenging.
>
> We instead use another visual-based MARL benchmark that provides easy access to a gym-like environment: **Google research football**. In this benchmark, each agent recieves RGB frames that contain global information of the environment and controls a player with a high-level action space (moving towards different directions, passing, shooting, etc). Since the observation is not local, it does not fit the setting of a decentralized POMDP. However, this is the closest setting we could find to test our method. We are working on running our method and baselines in this environment, and will upload them once we have the results.
>
> Regarding the comparison to the state-based setting, we also find it interesting to see how our method performs in this case. We therefore compared our method with MAPPO and MA2CL in Multi-agent Mujoco. In three Mujoco domains, MA2CL consistently outperforms MAPPO while still performing worse than reported in their original paper. MA-TRVAE outperforms MA2CL in two domains ((8x1 Ant and 6x1 Half$\_$cheetah) by a large margin and performs slightly worse than MAPPO in the 3x1 hopper. Nevertheless, we observe a smaller variance of MA-TRVAE in the hopper domain. Thus, overall, we see that also in the state-based setting, especially for high-dimensional systems, the trust region approach provides a margin over existing approaches. We will upload the results together with the ones for Google research football in the next week.
>
> Regarding EMA as an alternative to the trust-region constraint, we had not previously considered it. The brief answer is that *the stability in parameter space does not lead to stability in the function space*. While EMA can smooth parameter updates, it does not explicitly control how much the latent distributions change between updates. Because VAE encoders are highly nonlinear and their parameters are high-dimensional, even small parameter shifts can still result in large changes in the latent representations. In contrast, our trust-region constraint regularizes the latent outputs directly, ensuring bounded changes in representation space and providing more targeted stabilization for the policy learning in the MARL setting.

---

> > ### Comment · Reviewer_Enue · 2025-11-22
> >
> > Thank you for the response! I will continue considering all of that throughout the rest of the review process.  And slightly more details about my comment:
> >
> > > the MA2CL results reported in the paper under submission seem significantly worse than reported
> >
> > For example, MA2CL in the leader-follower with RPM reward in Table 5 in the submitted paper diverges to a reward of -80 while the submitted method converged to almost -10. However in [Figure 3](https://arxiv.org/pdf/2306.02006) of the MA2CL paper, MA2CL does not diverge like this and also attains a reward of almost -10.

---

> > > ### Author Response · Authors · 2025-12-03
> > >
> > > We thank the reviewer for the response. We have uploaded additional results in the updated paper as we stated in the general response.
> > >
> > > Regarding the MA2CL performance, we agree that the performance is worse than was reported. However, we conducted experiments for MA2CL strictly following the original paper, and we do not know the reason for the degradation in their performance. Also, we would like to point out that **the MA2CL we compare with is not the MA2CL based on MAT**, which is the MA2CL in Figure 3 of the MA2CL paper. To ensure fair comparison, we instead compare with MAPPO + MA2CL, which is based on a CNN encoder, since our MA-TRVAE is based on MAPPO and a CNN encoder. In Figure 11 of the MA2CL paper, MAPPO + MA2CL seems to converge to a reward lower than -10 for the leader-follower with RPM action mode after 1M steps.
> > >
> > > Nevertheless, we realized that MA2CL may not be a reliable SOTA method to compare against, so **we added MAT (Multi-agent Transformer) as a baseline**. The results show that MAPPO + MA-TRVAE performs better than MAT under PID action mode but slightly worse under RPM and DYN action modes (see Fig. 18). In all tasks, MAPPO + MA-TRVAE shows less variance, showing the stability of our method. In addition, we found MAT's performance and stability drop significantly when we scale up the number of agents in MAQC, similar to MA2CL, while our MAPPO + MA-TRVAE approach can scale to a higher number of agents without performance degradation (see Fig. 19).
> > >
> > >
> > > Finally, we observed that the performance of baseline methods including MAPPO and MAT is better than the one reported in MA2CL paper (see Figure 3 and 11 in MA2CL paper for comparison), which we also don't understand why, since we use the repository from the MA2CL paper to run MAPPO and MAT, and use the same hyperparameters as reported in their paper.

---

### Author Response · Authors · 2025-12-03
**General response**

We thank the Area Chair for overseeing the review process and the reviewers for their thoughtful, detailed, and constructive feedback. Importantly, **all four reviewers responded positively to our method and were open to further discussion**, with some explicitly indicating willingness to raise their scores pending our additional experiments. Reviewers consistently appreciate the clarity of the
paper, the significance of identifying representation drift, and the simplicity and empirical value
of the proposed trust-region constraint. To contextualize these contributions, we briefly summarize our method: MA-TRVAE introduces a trust-region-regularized VAE that stabilizes visual representations by constraining changes in latent posteriors across updates, mitigating representation drift that can destabilize MARL training. Integrated with MAPPO, our approach leads to more stable learning, greater sample efficiency, and better scalability than reconstruction- or contrastive-based baselines (MA2CL). The reviewer's comments helped us further improve the submission, and we summarize below how we addressed the major concerns through additional experiments (we show them in section H in the updated paper with red text).

**Limited Experiment Scope**

 All reviewers raised concerns about the evaluation being limited to the MAQC quadcopter benchmark. Reviewer Enue acknowledged the difficulty in finding a visual-based MARL benchmark. To address this:
-  We extended our evaluation to a state-based MARL benchmark, **Multi-agent Mujoco**, covering five domains (see Fig.13 for results).
   - 6×1 HalfCheetah, 6x1 Walker, 8×1 Ant, and 10x2 Swimmer: MA-TRVAE outperforms MA2CL and MAPPO by a large margin,
   - 3×1 Hopper: Slightly lower final return but significantly lower variance than baselines.
- We extended our evaluation to another visual-based MARL benchmark, **Google Research Football** (see Fig.14 for the result).
   - This benchmark does not provide local observations (ego-centric view) for each agent, but agents share a global observation (the frame of the whole screen) across all agents. Thus, the benchmark does not exactly fit our method.
   - MA-TRVAE outperforms MAPPO and slightly underperforms MA2CL.
   - We see potential to further improve the performance through hyperparameter tuning (see section H.2 for more details), which we couldn't do in this rebuttal period due to lack of time and resources. However, even without hyperparameter tuning, we see that we can outperform a state-of-the-art method and are almost on par with another one, which we would consider a strong result.

These results demonstrate that MA-TRVAE generalizes beyond the quadcopter domain and visual observation. Furthermore,  we emphasize that **we used the same hyperparameters from MAQC for MA-TRVAE across all additional benchmarks**, given the limited time and computing resources. Due to the large differences between these benchmarks, we expect better performance with adjusted hyperparameters ($e.g., \beta_2$, VAE learning rate). See sections H.1 and H.2 for more details.

**Limited Baseline Coverage**

Reviewer Enue pointed out that MA2CL underperforms compared to their reported results, and reviewer 16Nd suggested evaluating against transformer-based or MAE-based baselines.
- **We therefore added comparisons to MAT**, a strong transformer-based MARL method. We found MAT is a stronger baseline than MA2CL in the MAQC benchmark. Nevertheless, the results show
    - MA-TRVAE has better performance than MAT in PID control mode, and slightly lower performance in two lower-level control modes (see Fig.18 for results),
    - significantly better scaling behavior as the number of agents increases. While MAT's performance degrades as we increase the number of agents, MA-TRVAE maintains a consistent performance (see Fig. 19 for results).
- We clarified that MAE-based methods (MA²E, MA2RL) address partial observability in state-based MARL rather than learn a good representation for visual observation. Their architectures do not trivially extend to visual-based MARL, making them conceptually complementary rather than direct baselines.

These comparisons strengthen the empirical positioning of MA-TRVAE among other strong methods. See section H.4 for more details.

**Sensitivity to $\beta_2$ (Trust-Region Constriaint Strength)**

Most reviewers emphasized the importance of analyzing hyperparameter sensitivity. We therefore conducted an study of $\beta_2$ effect over values from 1e-1 to 1e-8 (see Fig.17 for results):
- Large $\beta_2$ → over-constrained latent space → stability but poor exploration → weaker MARL performance.
- Small $\beta_2$ → flexible latent space → richer exploration → better performance.

This confirms our intuition: stability is necessary, but over-regularization hurts the expressivity in the latent space. See section H.3 for more details.

---

> ### Author Response · Authors · 2025-12-03
>
> **Robustness to Observation Noise**
>
> Reviewer 16Nd pointed out that a deeper analysis of more challenging noise conditions is missing for justifying our robustness claim.
> - Our experiments demonstrate empirical robustness under additive Gaussian noise.
> - Following Reviewer 16Nd’s suggestion, we have revised our claims accordingly to avoid overstating robustness.
>
> **We believe that the additional experiments and our refined claim for robustness have addressed all reviwers' concerns and expect reviewers to raise their scores** (reviewer 16Nd has raised the score to 6 before we upload the additional results).
>
>
> Apart from the concerns, several questions were raised across the reviews. We addressed all of them. Below, we summarize some key questions:
> - **How is the trust-region constraint compared to simpler stabilization mechanisms such as EMA or temporal latent smoothing?**
>
>
>    We explained that such techniques stabilize the parameter space of the encoder or time-adjacent latents, whereas MA‑TRVAE stabilizes the encoder's output space directly, which is more relevant for MARL;
>
> - **Can the trust‑region idea be applied to contrastive or predictive representation learning approaches such as MA2CL?**
>
>
>    We clarified that the mechanism can conceptually extend to methods lacking strong latent space regularization, though it is most beneficial for reconstruction-based encoders, where a less regularized latent space leads to better MARL performance (see Fig. 2).
>
> - **Whether alternative trust-region penalties (e.g., symmetric KL, Wasserstein, or JS divergence) were considered?**
>
>
>    We explained that we deliberately use KL for both the VAE prior and the trust-region term to keep the objectives consistent and avoid potentially conflicting gradients.
>
> We sincerely appreciate the reviewers’ positive evaluations and constructive input, which have substantially improved the final paper. We will further fine-tune our method in these additional benchmarks and integrate results into the final version of the paper. We hope these updates address the remaining concerns and demonstrate that MA-TRVAE is an efficient, scalable, and general approach to stabilizing representation learning in vision-based multi-agent RL.

---

### Meta-Review · Area_Chair_YuVv · 2025-12-28

**Summary:**

This paper explores efficient and scalable vision-based multi-agent reinforcement learning (MARL) methods, addressing a significant challenge related to sample efficiency. To solve this problem, the authors propose the Multi-agent Trust Region Variational Autoencoder (MA-TRVAE) framework. The authors aim to improve sample efficiency and stability in MARL tasks and experimentally demonstrate the effectiveness of the proposed method in vision-based MARL. Reviewers raised numerous questions regarding the method's experiments, design motivation, and especially the method settings and comparisons. The authors partially addressed these concerns in their response, and some reviewers expressed a positive attitude compared to the initial system score. See the detailed analysis below for further details.

**Reviewer Concerns:**

+ Reviewer Enue acknowledged the experimental presentation and expression to some extent. However, the reviewer expressed concerns about the experimental design, particularly regarding beta-VAE representations and trust-region stabilization. They were particularly concerned about the quadcopter settings, including the lack of Starcraft and Mujoco environments, ultimately awarding a low score of 2. In the discussion and response section, the reviewer remained concerned about the reported effectiveness of the MA2CL method. Although the authors stated, "We do not know the reason for the degradation in their performance," they still provided considerable explanation, which is commendable. The area chair understands the difficulties in reproducibility or code expression.

+ Reviewer 16Nd, while generally negative about the paper, affirmed its methodological expression and clarity. Effective discussions took place between the authors and reviewers, including regarding the expression of key descriptions. However, considering that experiments have not yet been conducted, the reviewer remains positive about these potential experiments. The area chair further confirmed the updated paper and partially approved these revisions.

+ Reviewer cW5t raised specific concerns about the narrow empirical scope, particularly the limitation to the MAQC drone environment. However, due to time and resource constraints, the authors were unable to complete sufficient and comprehensive experiments. The authors' current method achieved promising results in some settings, but problems remain in many environments.

+ Furthermore, reviewer me5b expressed concern about the paper's technical innovation, the experimental evaluation dataset and scope, and the lack of theoretical basis.

**Reviewer Scores:**

In summary, this paper initially received 3 negative scores and 1 positive score, with one negative score being low and highly confident. Throughout the discussions, a high level of concern remained, particularly regarding the experimental aspects and the comparison with the MA2CL method. However, it's worth noting that reviewer 16Nd anticipated future improvements would be effective and was willing to believe in the authors' related improvements. This placed the paper at a borderline score that was generally negative but open to discussion. The area chair considered the questions regarding the experiments and innovativeness, and acknowledged the authors' positive contributions, but still believed the paper had significant room for improvement, especially requiring some experimental revisions, and that a complete re-review at this stage would be difficult.

---

### Decision · Program_Chairs · 2026-01-26

Reject